# Gut microbiota-derived butyrate primes systemic immunity in honey bees by mediating lipid metabolic reprogramming

Jiaming Liu [ID] , Yashuai Wu [ID], Zhenfang Li, Junbo Tang, Xin Zhou [ID] & Shiqi Luo [ID] [✉]

The gut microbiota plays a crucial role in insect immune priming, inducing enhanced immune response that functionally resembles acquired immunity confined to vertebrates. While gut microbiota mediates systemic immune activation in insect hemolymph, the mechanisms underlying remote immunoregulation remain largely unknown. Here we use the honey bee gut microbiota as a model, we identify butyrate as a key microbial metabolite coordinating immune-metabolic crosstalk. Butyrate supplementation restores immune competence in germ-free bees, mirroring the protective effects of microbiota-colonized individuals. Butyrate orchestrates lipid metabolic reprogramming in the fat body by activating glycerolipid and arachidonic acid metabolism through activating the G-protein coupled receptor 41 while inhibiting histone deacetylases. These changes in-turn upregulate prostaglandin $E_2$ biosynthesis, which is essential for humoral and cellular immune activation. These findings unravel how the intricate integration of immune and metabolic systems in honey bees is driven by gut-host interactions.

Although insects lack acquired immunity, their innate immune system can develop enhanced defensive capacity through "immune priming"[1]. This process enhances immune competence of the host after initial pathogen exposure, enabling more rapid and effective resistance and tolerance to subsequent infections[1-3]. Besides pathogens and their microbe-associated molecular patterns, immune priming can also be induced by nonpathogenic bacteria or symbionts, stimulating the immune system to boost host defenses against subsequent pathogenic challenges[3,4]. Studies have shown that insect gut microbiota can prime both local gut immunity and systemic immunity occurring in the hemolymph. The systemic immunity involves cellular immunity mediated by hemocytes, and humoral immunity mediated by effector molecules secreted from hemocytes and the fat body[5]. While the role of gut microbiota in regulating local gut immunity has been confirmed across diverse insect taxa[6-13], its effects in systemic immunomodulation have only been studied in limited species, including mosquitoes[14], honey bees[15,16], red palm weevils[17], and bean bugs[18,19]. Despite that

systemic immunity is essential for hemolymph pathogen clearance in insects, how gut microbiota regulates this process remains poorly understood.

The western honey bee (*Apis mellifera*) serves as important pollinators and a good model system for investigating host-microbe symbiosis[20,21]. Unlike many other insects, adult worker honey bees maintain a simple and stable gut microbiota composition, typically harboring $10^8$-$10^9$ bacterial cells dominated by five core lineages: *Gilliamella*, *Snodgrassella*, *Bifidobacterium*, *Bombilactobacillus*, and *Lactobacillus*[21,22]. These gut symbionts play crucial roles in immunoregulation, including local gut immune priming through increased expression of antimicrobial peptides (AMPs) and other immune-related genes[10,15,23,24]. They also induce systemic immune activation by elevating apidaecin levels in the hemolymph[15] and upregulating AMP gene expression in the fat body[16]. Notably, live *Snodgrassella alvi* colonization confers superior protection against pathogen infection compared to heat-killed bacteria and induces unique immune gene

State Key Laboratory of Agricultural and Forestry Biosecurity, MOA Key Lab of Pest Monitoring and Green Management, Department of Entomology, College of Plant Protection, China Agricultural University, Beijing, China. [✉]e-mail: shiqi_luo@cau.edu.cn

expression profiles in honey bees[16], suggesting bacterial metabolite-mediated immunoregulation. Microbial metabolite short-chain fatty acids (SCFAs), known for their role in activating systemic immunity in mammals[25], are also produced by honey bee gut microbiota[26,27]. But their potential role in the gut-host immune interaction in the honey bees remains unknown.

In this study, we investigated how gut microbiota primes honey bee systemic immunity. We challenged honey bees by injecting them with *Hafnia alvei*, a bacterium commonly detected in the hemolymph and guts of worker bees and known to cause septicemia and gut inflammation, thereby reducing overall survival[10,23,28,29]. Under *Hafnia* infection, we found that conventionally colonized bees (CV), which possess a complete and active gut community, exhibited higher survival rates than both bees colonized with heat-killed microbiota (HK) and germ-free bees (GF) lacking gut bacteria. This result demonstrates that the metabolic activity of live gut microbes is essential for enhancing honey bee survival against pathogenic infection. The CV bees showed obvious upregulated AMP gene expression and dorsal vessel-associated hemocyte aggregation in the fat body, which represent humoral and cellular immune response, respectively[30].

In the insect circulatory system, the dorsal vessel comprises an aorta in the thorax and a heart in the abdomen. The periostial regions surrounding the heart valves (ostia) are zones of intense hemolymph flow, making them critical sites for pathogen clearance[30]. Insect hemocytes exist in both sessile and circulating states. Following infection, resident sessile hemocytes in periostial regions become activated while circulating hemocytes migrate to these regions, resulting in localized aggregation and enhanced phagocytosis[31]. This periostial immune response represents a conserved and functionally significant defense mechanism across diverse insect taxa[32]. The extent of periostial hemocyte aggregation has been quantitatively demonstrated to scale in a time- and dose-dependent manner with infection intensity[33].

Next, we identify butyrate as the key microbial-derived SCFA to promote lipid metabolic reprogramming in the honey bee fat body, leading to increased production of prostaglandin $E_2$ (PGE$_2$) and subsequent activation of the systemic immune response. This study elucidates a microbiota-metabolite-immune axis in honey bees, where butyrate-mediated metabolic crosstalk regulates systemic immune priming against pathogenic challenges. These findings advance our understanding of the interplay between microbial metabolites and host immunity, and offer a foundation for designing probiotic or metabolite-based strategies to improve pollinator health.

## Results

### Live gut bacteria prime systemic immunity in honey bee host

To investigate the role of gut bacteria in systemic immune priming in honey bees, we compared bee survival rates under *Hafnia* infection for workers treated with varied gut bacterial conditions: germ-free (GF), heat-killed bacteria colonized (HK), and conventionally colonized (CV) (Fig. 1a and Supplementary Fig. 1a). The colonization of conventional gut bacteria (gut homogenate) significantly increased the survival rate of honey bees under pathogen infection, when compared with the GF group. Active biological components associated with living bacteria had likely played a significant role in improving host survival because bees inoculated by conventional gut bacteria showed notably higher survival rate than those by heat-killed bacteria (Fig. 1b and Supplementary Fig. 1b).

The temporal analysis of AMP expression revealed distinct patterns across groups and time points (Fig. 1c and Supplementary Fig. 1c). When comparing groups at each time point, significant upregulation relative to GF controls was observed at 3 hours post-infection (hpi) for *Apidaecin* in the CV group, *Defensin-1* in the HK group, and *Hymenoptaecin* in both the HK and CV groups. By 12 hpi, *Apidaecin* and *Hymenoptaecin* were elevated in the HK group, and by 24 hpi, *Abaecin*

was specifically upregulated in the HK group. Analysis of expression changes over time within each group (relative to 3 hpi levels) showed that *Apidaecin* in the HK group and *Hymenoptaecin* in the HK and CV groups were significantly upregulated as early as 12 hpi. By 24 hpi, *Apidaecin*, *Defensin-1*, and *Hymenoptaecin* were significantly upregulated in all three groups, and *Abaecin* was upregulated in the HK group. In addition, the CV group exhibited significant upregulation of Toll (*Toll*, *Cactus2*, *Dorsal*) and IMD (*Dredd*, *Relish*) pathway genes at 3 hpi, and no significant changes were detected in the HK group (Supplementary Fig. 1d).

Besides AMP expression, we quantified the conserved immune response of periostial hemocyte aggregation. This was done by measuring the periostial hemocyte number using CM-DiI, a well-established dye that specifically labels phagocytic hemocytes without staining non-hemocyte tissues, as validated in diverse insect species[32,34]. Remarkably, following *Hafnia* infection, a notable aggregation of hemocytes was also observed at the dorsal vessel of honey bees in the CV group at 3 and 6 h after infection (Fig. 1d, e). These hemocytes were engaged in phagocytosis of *Hafnia* (Supplementary Fig. 1e). At 3 h post-infection, the periostial hemocyte number (quantified as the percentage of CM-DiI-positive fluorescent area relative to the total image area) in the CV group was approximately 2.6 times greater than that in the GF group and 2.4 times greater than that in the HK group (Fig. 1f). Hemocyte aggregation in the CV group was most prominent at 3-6 hpi and subsequently decreased, whereas no significant temporal changes occurred in the GF and HK groups (Fig. 1f and Supplementary Figs. 2, 3). These results suggest that honey bees bearing natural gut microbiota display increased resistance to pathogen infection compared to those deprived of gut bacteria or those inoculated with dead microbes. The CV group has exhibited stronger systemic humoral and cellular immune responses at the initial stage of pathogen infection, indicating that specific metabolites produced by live gut bacteria may have primed the systemic immunity in the honey bee.

### Butyrate derived from gut bacteria primes systemic immunity

The microbial-derived SCFAs activate systemic immunity in mammals[25]. In honey bees, acetate is most abundant in the hindgut of CV bees (>100 mM), while butyrate shows the highest level in hemolymph (22.8 mM)[26]. These SCFAs have recognized immune roles: acetate activates gut immunity in *Drosophila*[9], and butyrate increases immune-related gene expression in honey bees[35]. We applied 10 mM of acetate or butyrate to the bees to examine their role in increasing bee immune responses observed in this study. The metabolite concentration was chosen in approximation of natural conditions detected in honey bee hemolymph, which also showed no detectable effects on bee survival in our examination (Supplementary Fig. 4a).

In evaluations of SCFA effects on bee immune responses, we employed honey bees inoculated with a set of five core bacteria mixed at equal numbers (CL group) as the positive control, as opposed to bees inoculated with gut homogenates (CV bees). Natural bee guts exhibit variation in bacterial composition[36,37]. In practice, a balanced mixture with equal amounts of the core bacteria species serves as a close approximation to natural conditions while minimizing potential pathogenic effects[27,38]. Specifically, the CL bees were inoculated with a consortium consisting of an equal amount of five bacterial strains representing the five core gut bacterial taxa (*Gilliamella*, *Snodgrassella*, *Bifidobacterium*, *Bomilactobacillus*, and *Lactobacillus*). The abundances of each core bacterium in CL-treated bees were similar to those in the CV group[27]. We showed that germ-free bees orally supplemented with 10 mM butyrate exhibited a significantly higher survival rate following *Hafnia* infection compared to the GF group, albeit still lower than that of the CL group (Fig. 2a and Supplementary Fig. 4b). In contrast, acetate supplement did not show enhancement in survival rates (Fig. 2a and Supplementary Fig. 4b). Further analysis revealed

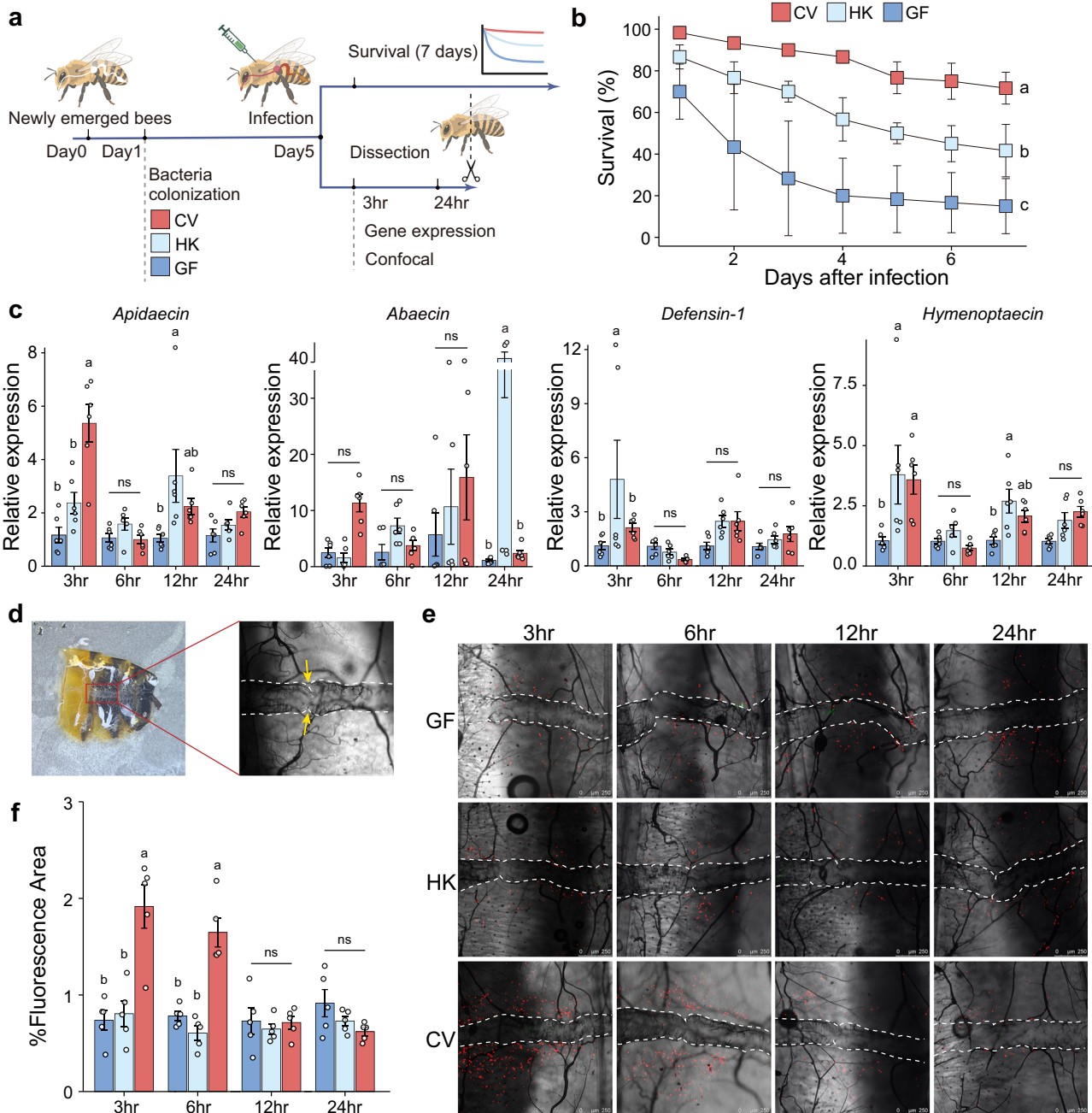

**Fig. 1 | Live gut bacteria prime systemic humoral and cellular immunity in honey bees. a** Experimental design for (**b**–**e**). Newly emerged worker bees were divided into GF (germ-free), HK (inoculated with heat-killed gut homogenate), and CV (inoculated with nurse bee gut homogenate) groups. Gut bacterial inoculation was conducted within one day after emergence. Bees were maintained for 5 days prior to *Hafnia* injection, and separate groups of bees were used for the survival assay, gene expression analysis, and confocal microscopy. Created in BioRender. Liu, L. (2025) https://BioRender.com/en83bag. **b** Survival rates for GF, HK, and CV bees post *Hafnia* injection. Bars represent mean ± s.d. from three biological replicates (20 individuals in each replicate). The results are from one of two independent experiments (Supplementary Fig. 1b for replicate data). A two-sided Mantel-Cox test was used. Different letters above bars indicate statistically significant differences among treatments ($P < 0.05$). **c** Antimicrobial peptide (AMP) gene expression in fat bodies of GF, HK, and CV bees, measured by qPCR (SYBR Green, *Actin*-normalized). Bars represent mean ± s.e.m. $n = 6$ individuals for each group from 3 cup cages. **d** Ventral view of bee abdominal tergum showing the dorsal vessel. Yellow arrows indicate the ostia. **e** Hemocyte aggregation post-GFP-*Hafnia* injection (green). Hemocytes were stained with CM-DiI (red), and dorsal vessels were outlined by white dashed lines. Scale bars: 250 μm. Representative images from one of two independent experiments (Supplementary Fig. S2–3 for replicate data). **f** Quantitative analysis of CM-DiI staining in hemocytes (see Fig. 1e and Supplementary Fig. 2). Bars represent mean ± s.e.m. $n = 5$ individuals per group from 3 cup cages. (**c**, **f**) Two-way ANOVA with Tukey's multiple comparisons test. Different letters above bars indicate statistically significant differences among treatments ($P < 0.05$). ns: not significant. All *P* values are listed in Supplementary Dataset 5. Source data are provided as a Source Data file.

that the colonization of each individual bacterium or their mixed consortium had all notably increased butyrate production in honey bee abdomen, except for *Bombilactobacillus*. The highest butyrate levels were found in *Bifidobacterium* colonized and CL groups (Fig. 2b).

*Hafnia* exposed to acetate or butyrate supplemented in the culture medium did not show changes in growth rate, indicating that these SCFAs did not directly affect the bacterial pathogen (Supplementary Fig. 4c, d). These findings suggest that butyrate may have

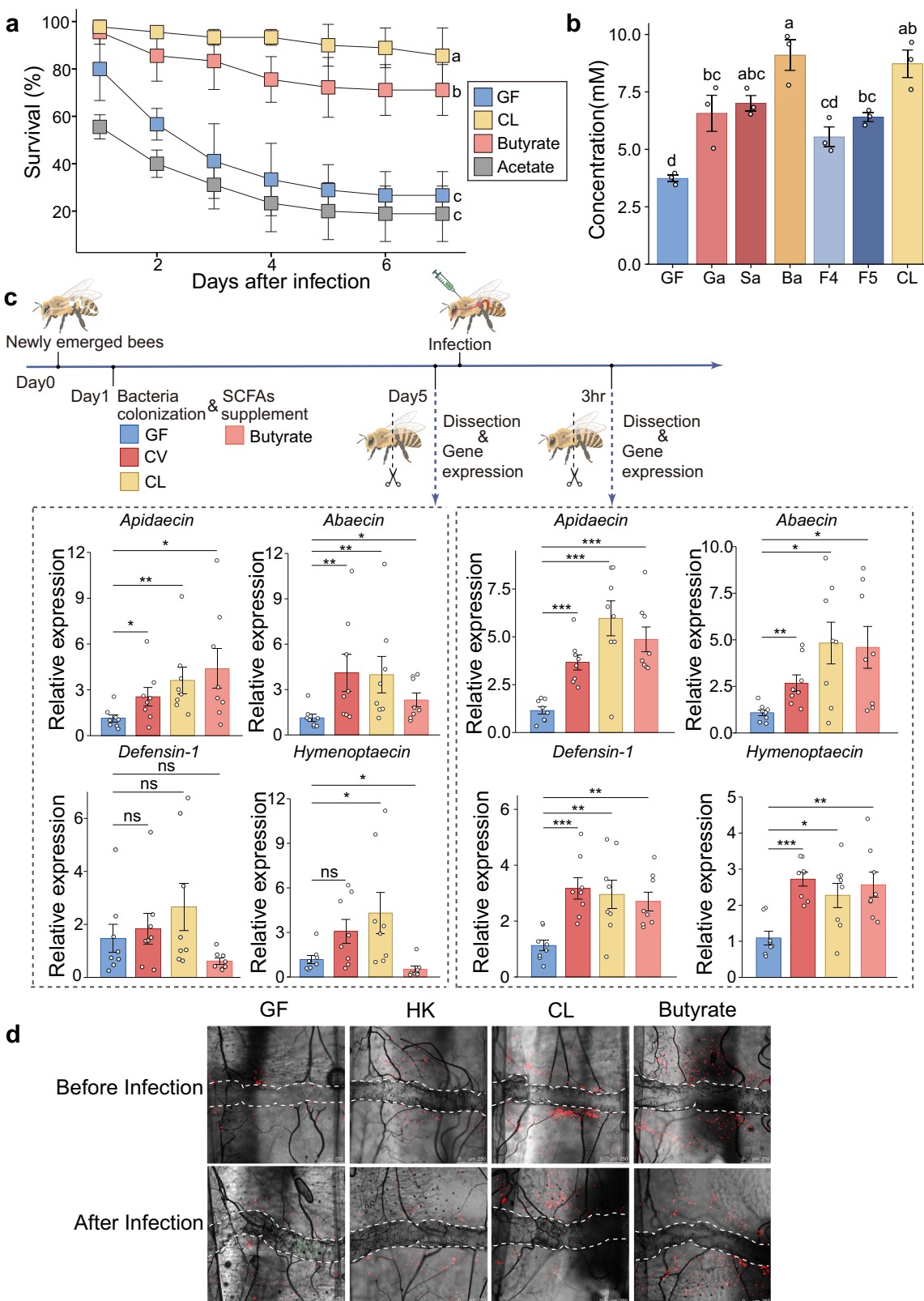

triggered host resistance to *Hafnia*, likely through activating host immune responses. Prior to *Hafnia* infection, the CV, CL and butyrate-supplemented groups already showed significantly increased expression levels of *Apidaecin* and *Abaecin* in the fat body compared to the GF group (Fig. 2c). *Hafnia* infection further elevated expressions of *Apidaecin*, *Abaecin*, *Hymenoptaecin*, and *Defensin-1* in the fat body of bees from the CV, CL and butyrate groups (Fig. 2c). In addition to the

increase of AMP expressions, the CL and butyrate groups also showed significantly higher hemocyte aggregation accompanying bacterial inoculation or butyrate feeding (Fig. 2d and Supplementary Figs. 5, 6), which was further elevated after *Hafnia* infection. In contrast, colonization with heat-killed gut bacteria showed no impacts on hemocyte aggregation, with or without pathogen infection (Fig. 2d and Supplementary Figs. 5, 6). In summary, the bacterial fermentation product

**Fig. 2 | Butyrate produced by honey bee gut bacteria primes host systemic immunity. a** Survival rates of GF, CL (inoculated with a set of five core bacteria mixed at equal numbers), acetate-supplemented, and butyrate-supplemented bees following *Hafnia* injection. Bars represent mean ± s.d. *n* = 3 replicates for each group (30 individuals per replicate). The results are from one of two independent experiments (Supplementary Fig. 4b for replicate data). A two-sided Mantel-Cox test was used. Different letters above bars indicate statistically significant differences among treatments (*P* < 0.05). **b** Abdominal butyrate concentrations for GF, CL, and single bacterium-inoculated bees 5 days post-colonization. Ga: *Gilliamella*; Sa: *Snodgrassella*; Ba: *Bifidobacterium*; F4: *Bombilactobacillus*; F5: *Lactobacillus*. The isolates for each strain are listed in Supplementary Table 1. Bars represent mean ± s.e.m. *n* = 3 replicates for each group, each replicate pooled from 3 individuals from 3 cup cages. The experiment was repeated twice with similar results. One-way ANOVA with the Tukey post hoc test. Different letters above bars indicate statistically significant differences among treatments (*P* < 0.05). **c** AMP expression in the fat bodies of GF, CV, CL, and butyrate-supplemented bees, before *Hafnia* infection and 3 h after infection (GF as control). Bars represent mean ± s.e.m. *n* = 8 individuals for each group from 3 cup cages. Gene expression was quantified using qPCR as shown in Fig. 1. Two-sided Mann–Whitney test and two-sided unpaired Student's *t* test, depending on data normality. ns: not significant, \**P* < 0.05, \*\**P* < 0.01, and \*\*\**P* < 0.001. Created in BioRender. Liu, L. (2025) https://BioRender.com/hdvmsxq. **d** CM-DiI-stained hemocytes (red) of GF, HK, CL, and butyrate-supplemented bees before infection and at 3 h post-GFP-*Hafnia* infection (green). Dorsal vessels were outlined with white dashed lines. Scale bars: 250 μm. Representative images from one of two independent experiments (see Supplementary Figs. 5, 6 for replicate data). All *P* values are listed in Supplementary Dataset 5. Source data are provided as a Source Data file.

butyrate notably up-regulates the expression of honey bee AMP genes and promotes hemocyte aggregation, which jointly prime systemic immunity of the host, increasing its resistance to pathogen.

## Butyrate enhances expression of honey bee genes related to lipid and arachidonic acid metabolism, leading to lipid breakdown

Significant differences in gene expression were observed in the fat body and hindgut between GF bees and those supplemented with butyrate (Fig. 3a). More differentially expressed genes (DEGs) were found in the fat body between the two groups, where the butyrate-supplemented group showed up-regulation in 451 genes and down-regulation in 454 genes (Supplementary Fig. 7a and Supplementary Dataset 1). In the hindgut, the butyrate-supplemented group exhibited up-regulation in 117 genes and down-regulation in 72 genes (Supplementary Fig. 7b and Supplementary Dataset 2).

The DEGs in the fat body were enriched in pathways including amino acid metabolism, carbohydrate metabolism and lipid metabolism (Fig. 3b). On the other hand, the DEGs in the hindgut were mainly enriched in the lipid metabolism pathway (Fig. 3b). Notably, both the fat body and hindgut DEGs were significantly enriched in genes related to lipid metabolism. Analysis of DEGs in the fat body showed enrichment in fatty acid degradation and glycerolipid metabolism pathways. Similarly, in the hindgut, butyrate supplementation led to notable up-regulation of *Gpdh*, *LOC550970*, *LOC408817*, *LOC102656448*, *Pla2*, and *LOC409307*, which are involved in glycerophospholipid metabolism and arachidonic acid metabolism pathways (Supplementary Fig. 7c).

Both transcriptome and qPCR results showed a significant up-regulation in genes associated with lipid metabolism in the honey bee fat body following butyrate supplementation, including *LOC408559* (retinal dehydrogenase 1), *LOC551968* (aldose reductase), *LOC409261* (glycerol kinase), *mino* (*LOC411724*, glycerol-3-phosphate acyltransferase mino), *LOC724951* (1-acyl-sn-glycerol-3-phosphate acyltransferase beta), *LOC724995* (1-acyl-sn-glycerol-3-phosphate acyltransferase alpha), *Lpin* (*LOC410201*, phosphatidate phosphatase), *LOC726880* (pancreatic lipase-related protein 2-like), *bbc* (*LOC552795*, choline/ethanolamine phosphotransferase), *Pla2* (*LOC406141*, phospholipase A2) (Fig. 3c, d). Specifically, lipid metabolism genes in the fat body are involved in glycerol synthesis and triacylglycerol (TAG) breakdown processes (Fig. 3c), which lead to diacylglycerol (DAG) production, an important substrate in immune-related arachidonic acid metabolism pathways in insects. Furthermore, two genes (*bbc* and *Pla2*) involved in arachidonic acid production from DAG are up-regulated in the butyrate supplementation group. Butyrate supplementation also reduced neutral lipid storage and lipid droplet size in the honey bee fat body (Fig. 3e and Supplementary Fig. 8), resulting in lowered TAG content (Fig. 3f). These results suggest that butyrate produced by gut bacteria can up-regulate the expression of host genes involved in lipid breakdown. We hypothesize that the breaking-down

of lipids may in-turn supply substrates for arachidonic acid metabolism and influence the balance between lipid storage and the production of immune-related compounds.

## Butyrate induces honey bees to produce prostaglandin E₂, priming systemic immunity

Metabolomic analysis of whole honey bee abdomens revealed significant alterations in 1189 metabolites (702 increased and 487 decreased) following butyrate supplementation (Fig. 4a and Supplementary Dataset 3). The arachidonic acid metabolic pathway was notably enriched in these changes (Supplementary Fig. 9a). Elevated levels of prostaglandin D₂ (PGD₂), prostaglandin E₂ (PGE₂), thromboxane B₂, and 20-carboxy-leukotriene B₄ were identified in the honey bee fat body after butyrate supplementation (Fig. 4b). Of particular interest, the PGE₂ plays a pivotal role in regulating insect immune processes[39-41]. Following gut bacterial colonization (CV and CL groups) and butyrate supplementation, a marked increase in PGE₂ level was detected in the abdomen, hindgut, and hemolymph of the honey bee (Fig. 4c and Supplementary Fig. 9b), aligning with our metabolomic results.

Among fat body DEGs induced by butyrate supplementation, *Pla2* is responsible for arachidonic acid production. This fatty acid is the precursor for prostaglandin biosynthesis[42]. Here, RNAi-mediated knockdown of *Pla2* reduced *Pla2* expression (Fig. 4d), which attenuated the promotion effect of butyrate on honey bee prostaglandin production, bringing down PGE₂ level to that of the GF group (Fig. 4e). These results suggest that butyrate enhances PGE₂ content by up-regulating lipid metabolism-related genes, leading to the break-down of lipids into DAG, which is then processed by PLA2 to produce PGE₂.

The CL group exhibited significantly higher PGE₂ levels compared to the GF controls, along with enhanced immune activation. We further examined the impact of PGE₂ on the systemic immune response of honey bees. PGE₂ injection to the GF honey bees increased their survival rates post-*Hafnia* infection, reaching a similar level as the CL group. On the other hand, the CL honey bees were also treated with acetylsalicylic acid (ASA, a prostaglandin synthesis inhibitor). To examine whether ASA exerts toxic effects on the bees, bees were fed different concentrations of ASA without bacterial infection, whereas 10 mM ASA showed no significant effect on survival (Supplementary Fig. 9c). However, following *Hafnia* infection, ASA-treated CL bees exhibited a pronounced reduction in survival, even lower than that of the GF group (Fig. 4f and Supplementary Fig. 9d). These results highlight the capacity of PGE₂ in systemic immunity priming in honey bees.

We further showed that PGE₂ evoked immune reactions of the honey bee the same way as the gut bacteria and butyrate did. Prior to *Hafnia* infection, supplementation of PGE₂ in the GF group increased the expression of *Apidaecin* and *Abaecin* (Fig. 4g). As expected, ASA treatment decreased the expression of *Apidaecin*, *Abaecin*, *Hymenoptaecin*, and *Defensin-1* in the CL group (Fig. 4g). After *Hafnia*

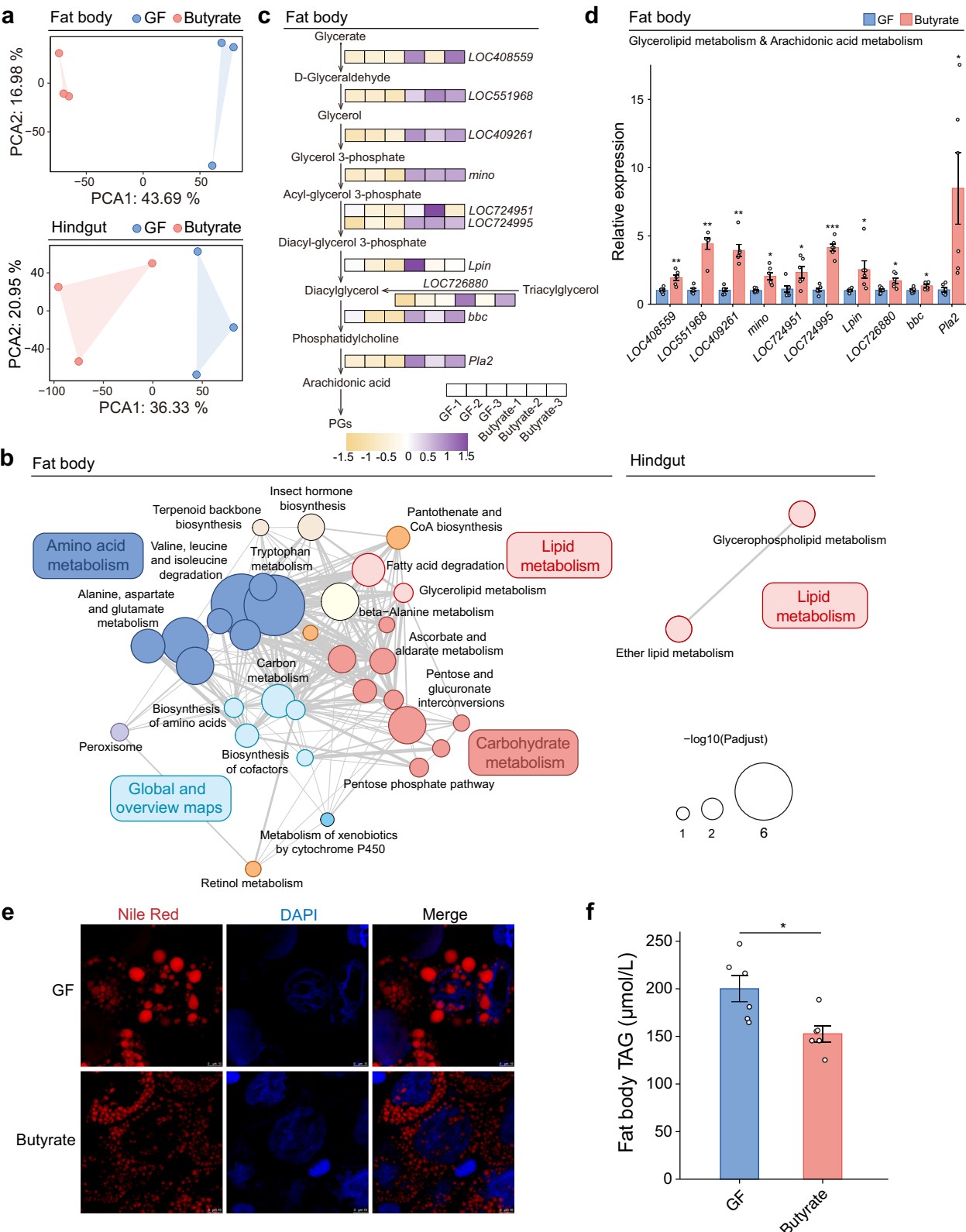

infection, PGE$_2$ supplementation in the GF group enhanced the expression of *Apidaecin*, *Abaecin*, *Hymenoptaecin*, and *Defensin-1*, while ASA treatment inhibited these AMPs in the CL group (Fig. 4g). PGE$_2$ supplementation also improved hemocyte aggregation in GF honey bees, while ASA treatment suppressed hemocyte aggregation in CL honey bees before and after *Hafnia* infection (Fig. 4h and

Supplementary Fig. 10-11). Combined, these results suggest that the gut microbiota induce the up-regulation of lipid metabolism genes through butyrate production, leading to PGE$_2$ synthesis. PGE$_2$ can enhance the expression of honey bee AMP genes and facilitate hemocyte aggregation, increasing honey bee resistance to pathogen infection.

**Fig. 3 | Butyrate enhances lipid breakdown in honey bees, producing compounds for arachidonic acid metabolism. a** Principal Component Analysis of transcriptomes of fat body and hindgut of GF and butyrate-supplemented bees (5 days post-supplementation). Each point represents one biological replicate, with three replicates per group. **b** Pathway enrichment for genes upregulated in the fat bodies and hindguts of butyrate-supplemented bees, using the GF group as the control. Circles representing pathways are clustered by functions, with edges showing the number of overlapping genes between pathways. **c** Heatmap of differentially expressed genes (DEGs, $P < 0.05$ and $|\log_2(\text{fold change})| \geq 1$) involved in lipid metabolism in the fat bodies. *LOC408559* (retinal dehydrogenase 1), *LOC551968* (aldose reductase), *LOC409261* (glycerol kinase), *mino* (*LOC411724*, glycerol-3-phosphate acyltransferase mino), *LOC724951* (1-acyl-sn-glycerol-3-phosphate acyltransferase beta), *LOC724995* (1-acyl-sn-glycerol-3-phosphate acyltransferase alpha), *Lpin* (*LOC410201*, phosphatidate phosphatase), *LOC726880* (pancreatic lipase-related protein 2-like), *bbc* (*LOC552795*, choline/ethanolamine phosphotransferase), *Pla2* (*LOC406141*, phospholipase A2). In these genes, *mino*, *LOC726880*, *bbc* ($P < 0.05$, fold change > 1.3); *LOC724951*, *Lpin* ($P = 0.05$ and fold change > 2). **d** qPCR validation of glycerolipid and arachidonic acid metabolism DEGs (GF as the control, *Actin*-normalized). Bars represent mean ± s.e.m. n = 6 individuals for each group from 3 cup cages. Two-sided Mann–Whitney test and two-sided unpaired Student's *t* test, depending on data normality. \**P* < 0.05, \*\**P* < 0.01, and \*\*\**P* < 0.001. All *P* values are listed in Supplementary Dataset 5. **e** Nile red staining of neutral lipids (red) in honey bee fat bodies on day 5 after butyrate supplementation, with cell nuclei stained with DAPI (blue). Representative images from one of two independent experiments (see Supplementary Fig. 8 for replicate data). Scale bars: 10 μm. **f** Total triacylglycerol (TAG) levels in the fat bodies. Bars represent mean ± s.e.m. n = 6 replicates for each group, with each replicate pooled from 3 individuals from 3 cup cages. $P = 0.0147$ from two-sided Student's *t* test. Source data are provided as a Source Data file.

## Butyrate regulates gene expression in honey bee fat body by triggering the GPR41 signaling pathway and by increasing histone acetylation levels

Butyrate can regulate gene expression via at least two known mechanisms: by activating the G-protein-coupled receptor GPR41 and downstream signaling pathway, or by inhibiting histone deacetylase (HDAC)[43,44]. To validate the regulatory role of butyrate through GPR41, GF honey bees were treated with butyrate alone, or in combination with β-hydroxybutyrate (SHB, a GPR41 antagonist)[45,46]. The potential toxicity of SHB was first assessed by feeding bees with varying concentrations of SHB. Results showed that neither 20 mM nor 50 mM SHB posed notable effects on bee survival (Supplementary Fig. 12a). All ten lipid metabolism-related genes, but *Lpin*, were reproducibly upregulated by butyrate (Fig. 5a). This upregulation was significantly suppressed for eight genes when butyrate was co-administered with SHB (Fig. 5a). Moreover, the inhibitory effect was more pronounced with 50 mM SHB than with 20 mM. In consistence, treatment with the GPR41-specific agonist AR420626 resulted in up-regulation for seven out of the ten genes involved in butyrate-induced lipid metabolism (Fig. 5b). These findings demonstrate that butyrate regulates lipid metabolism in honey bees via GPR41 activation, albeit not necessarily exclusively.

Further in vivo experiments demonstrated a significant reduction in HDAC enzyme activity in fat bodies upon butyrate supplementation (Fig. 5c and Supplementary 12b). The treatment of Trichostatin A (TSA), an HDAC inhibitor, brought up expression for seven genes related to butyrate-induced lipid metabolism (Fig. 5d). Given both AR420626 and TSA treatments have notably accelerated lipid droplet breakdown (Fig. 5e and Supplementary Figs. 13, 14), we infer that butyrate might regulate lipid break-down in honey bees via both GPR41 activation and histone acetylation modulation.

To validate the regulatory effects of butyrate on histone acetylation, we examined whether these acetylation modifications were present in DEGs related to lipid metabolism. Western blot analysis showed a significant increase in H3K27ac levels in honey bee fat body after butyrate supplementation (Fig. 6a and Supplementary Fig. 15a). CUT&Tag sequencing also revealed a change in the histone H3K27ac pattern in the fat body upon butyrate treatment (Fig. 6b). H3K27ac was enriched at transcription start site (TSS) regions in both GF and butyrate-treated groups, while the IgG control group showed minimal background signals (Fig. 6c). The H3K27ac modification was predominantly concentrated in promoter regions, accounting for 52% of total H3K27ac signals (Fig. 6d).

KEGG analysis of the differential peaks of H3K27ac modification revealed that the butyrate treatment resulted in up-regulations in tyrosine metabolism, glycerophospholipid metabolism, glycerolipid metabolism, and dorso-ventral axis formation pathways (Fig. 6e). In the glycerophospholipid metabolism pathway, genes such as *LOC724907*, *LOC724951*, *LOC724995*, and *LOC725668* exhibited significantly increased H3K27ac modification in the butyrate treated group (Fig. 6f). The expressions of *LOC724995* and *LOC724951* were also up-regulated after butyrate treatment (Fig. 3d). Particularly, H3K27ac peaks were notably enriched in the promoter regions of *LOC724951* and *LOC724907* (Fig. 6g). CHIP-qPCR analysis confirmed that butyrate notably increased H3K27ac modification levels in the promoter regions of *LOC724951* (Fig. 6h and Supplementary Fig. 15b) and *LOC724907* (Fig. 6i and Supplementary Fig. 15b). Both genes encode the 1-acyl-sn-glycerol-3-phosphate acyltransferase[47–49]. This enzyme catalyzes the conversion of acyl-glycerol 3-phosphate to diacylglycerol 3-phosphate, a process involved in the production of arachidonic acid and prostaglandin. These findings indicate that butyrate influences the balance between lipid metabolism and immune activation in the honey bee fat body through epigenetic modifications.

Based on the combined results, we conclude that the honey bee gut bacteria produce butyrate, which enters the bee fat body and increases the expression of genes related to lipid metabolism and arachidonic acid metabolism via two mechanisms (Fig. 6j). Firstly, butyrate binds to GPR41, activating its downstream signaling pathway. Secondly, butyrate inhibits HDAC enzyme activity, leading to an increase in H3K27ac modification in genes related to lipid metabolism (such as *LOC724951* and *LOC724907*) and arachidonic acid metabolism. These mechanisms collectively upregulate gene expression and promote $PGE_2$ production. Elevated $PGE_2$ in the fat body in turn up-regulates AMP gene expression and promotes hemocytes aggregation in the dorsal vessel, priming systemic immunity and enhancing honey bee defense against pathogens. Ultimately, gut microbiota-produced butyrate regulates honey bee lipid metabolism and maintains immune homeostasis of honey bees, significantly improving their systemic immune capacity against pathogens.

## Discussion

How gut symbiotic bacteria regulate host immune system is a central question in understanding host-microbe interactions. Although existing studies have demonstrated that gut symbionts can prime insect systemic immunity and enhance host resistance against pathogens, crucial details are missing: What is the key microbial metabolite that mediates host immune reaction? Which host signaling pathways are involved and how are they regulated? Through transcriptomics, metabolomics, RNAi, CUT&Tag, and in vivo experiments, our study reveals gut microbiota-derived butyrate regulates lipid metabolic reprogramming in the honey bee fat body to increase $PGE_2$ production, which primes systemic immunity and increases host survival under pathogen infections.

Previous studies have shown that systemic immunity in insects can be activated by constitutional components of the bacteria, e.g., cell wall peptidoglycan (PGN) from the bacterium *Erwinia carotovora carotovora* 15 (Ecc 15) associated with *Drosophila melanogaster*[50,51]. Such a mechanism may have also been involved in the honey bee

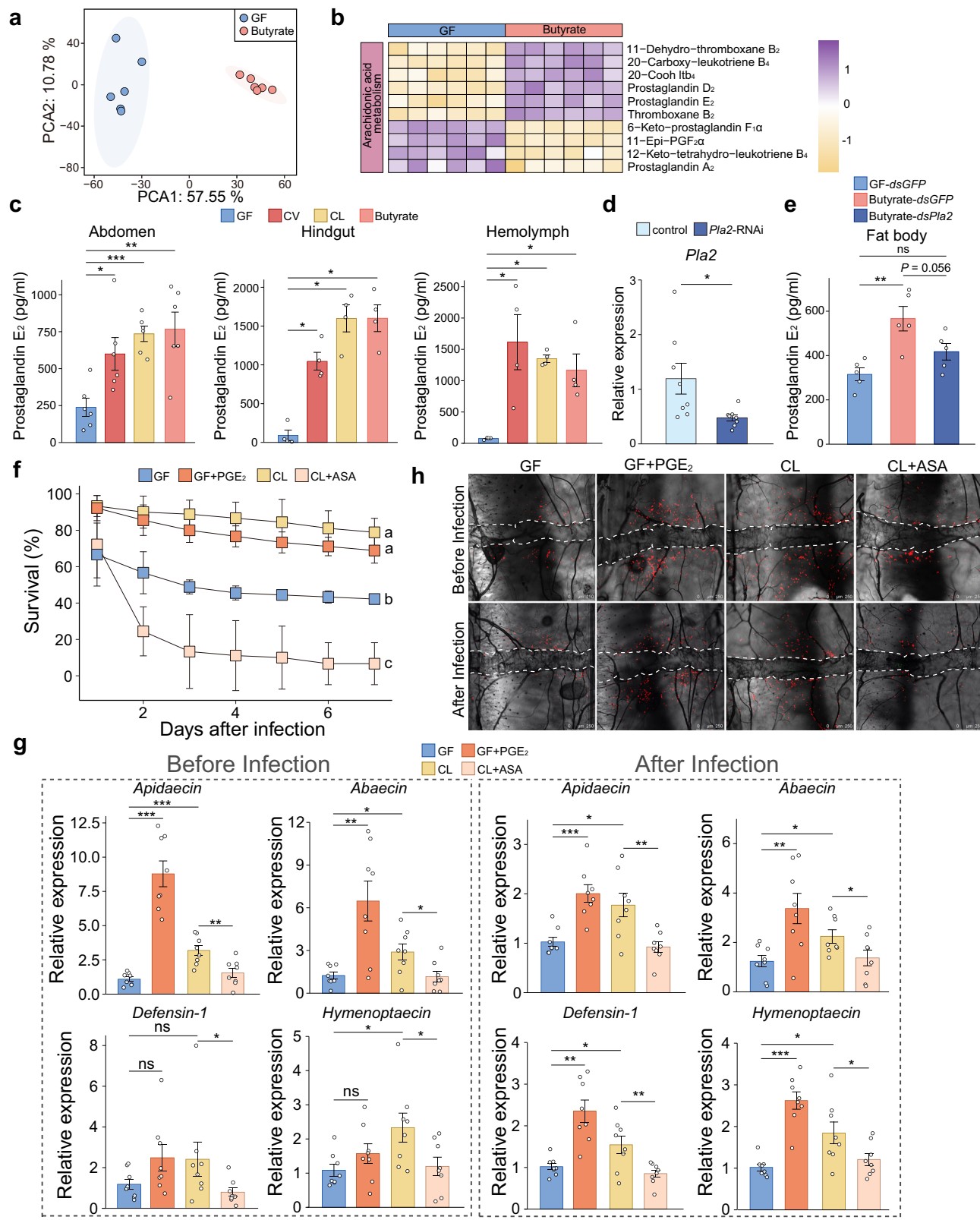

system here, which potentially explains why bees inoculated with dead bacteria (heat-killed, HK bees) still demonstrated enhanced survival under pathogen infection. Interestingly, although bees colonized with HK bacteria did not exhibit any change in periostial hemocyte aggregation, their immune gene expression profiles were significantly altered upon pathogenic infection. This suggests that the activation signals or thresholds for cellular and humoral immunity in insects may

be distinct. Although the temporal dynamics of these two immune responses were not entirely consistent, they act in concert in the systemic immune defense.

Our study further revealed that the systemic immunity of insect hosts can also be induced by metabolites produced by their gut microbiota. This mechanism enables bacterial regulation of the host without physical penetration of the host gut barrier, as demonstrated

**Fig. 4 | Honey bee prostaglandin production is stimulated by gut microbiota.**
**a** PCA of abdominal metabolomes from GF and butyrate-supplemented bees at day 5 post-supplementation (n = 6 per group, each point represents one biological replicate). **b** Heatmap of the discriminating metabolites in the arachidonic acid pathway in the abdomen of GF and butyrate-supplemented bees. **c** Prostaglandin E$_2$ (PGE$_2$) levels in abdomen, hindgut and hemolymph in GF, CV, CL, and butyrate-supplemented bees at day 5 post-colonization (abdomen: n = 6; hindgut and hemolymph: n = 4; each pooled from 5 individuals from 3 cup cages). One of two independent experiments (Supplementary Fig. 9b). Two-sided Mann–Whitney and two-sided unpaired Student's t test, depending on data normality. **d** Knockdown of fat body *Pla2* in GF bees following dsRNA injection (GFP dsRNA as control). n = 8 per group from 3 cup cages. P = 0.0394, two-sided Welch's t test. **e** Effect of *Pla2* RNAi on PGE$_2$ levels in fat bodies of GF and butyrate-supplemented bees (n = 5 replicates per group, each pooled from 5 individuals from 3 cup cages). P = 0.0039,

two-sided Student's t test. **f** Survival of GF, GF + PGE$_2$, CL, and CL+acetylsalicylic acid (ASA) bees post-*Hafnia* infection. Bars represent mean ± s.d. from three replicates per group (30 individuals per replicate) from one of two independent experiments (Supplementary Fig. 9d). Two-sided Mantel–Cox test; different letters indicate significant differences (P < 0.05). **g** AMP expression in fat bodies before and 3 h post-*Hafnia* infection (GF as control; n = 8 per group from 3 cup cages. Two-sided Mann–Whitney and two-sided unpaired Student's t test, depending on data normality. **h** In vivo hemocyte staining using CM-DiI (red) before infection (3 h after PGE$_2$ supplementation) and at 3 h post-infection with GFP-*Hafnia* (green). Dorsal vessels were outlined with white dotted lines. Scale bars: 250 μm. Representative images from two independent experiments (Supplementary Figs. 10, 11). In (**c**–**e**, **g**), bars represent mean ± s.e.m. *P < 0.05, **P < 0.01, and ***P < 0.001. All P values are listed in Supplementary Dataset 5. Source data are provided as a Source Data file.

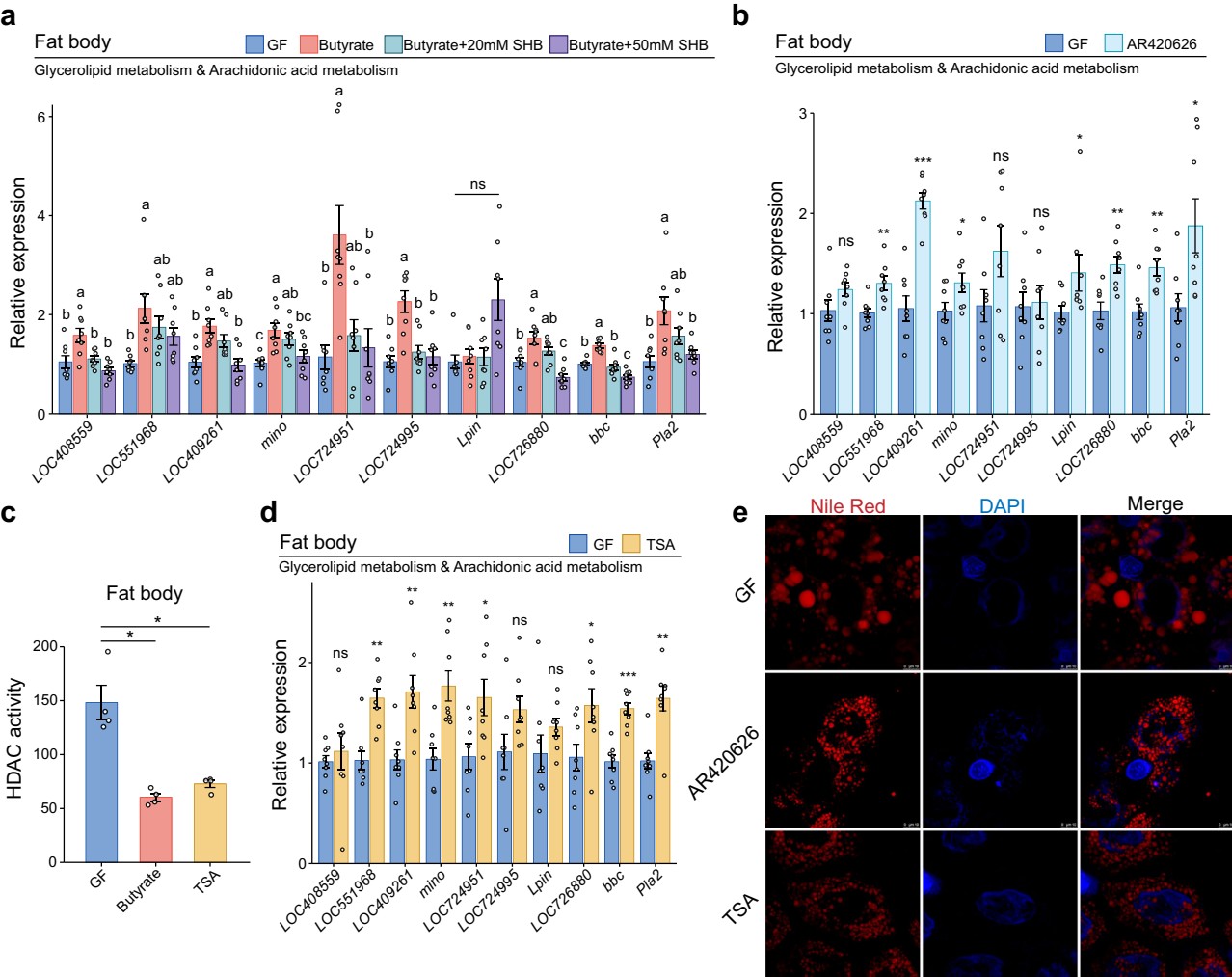

**Fig. 5 | Butyrate activates the GPCR pathway and regulates histone acetylation in honey bees, modulating the balance between lipid metabolism and immune activation.** **a** Expression of genes involved in glycerolipid metabolism and arachidonic acid metabolism in the bee fat body after β-hydroxybutyrate (SHB) supplementation (GF as control). n = 8 individuals for each group from 3 cup cages. One-way ANOVA with Tukey's test or Kruskal–Wallis with Dunn's test, depending on data normality. Different letters above bars indicate statistically significant differences among treatments (P < 0.05). **b** Expression of genes involved in glycerolipid metabolism and arachidonic acid metabolism in AR420626-treated groups (GF as control). n = 8 individuals for each group from 3 cup cages. **c** Fat body HDAC activity in GF, butyrate-supplemented and TSA-treated bees. Bars represent mean ± s.e.m. n = 4 replicates for each group, with each replicate pooled from 5 individuals. The results are from one of two independent experiments (Supplementary Fig. 12b

for replicate data). Two-sided Mann–Whitney test. *P < 0.05. **d** Expression of genes involved in glycerolipid metabolism and arachidonic acid metabolism in TSA-treated group (GF as control). n = 8 individuals for each group from 3 cup cages. In (**a**, **b**, **d**), Gene expression was quantified using qPCR, following protocols shown in Fig. 1. Genes are as described in Fig. 3c. Bars represent mean ± s.e.m. In (**b**, **d**), two-sided Mann–Whitney test and two-sided unpaired Student's t test, depending on data normality. ns: not significant, *P < 0.05, **P < 0.01, and ***P < 0.001. **e** Nile red staining of neutral lipids (red) in fat body of GF, AR420626- and TSA-treated bees. Cell nuclei are stained with DAPI (blue). Representative images from one of two independent experiments (see Supplementary Figs. 13, 14 for replicate data). Scale bars: 10 μm. All P values are listed in Supplementary Dataset 5. Source data are provided as a Source Data file.

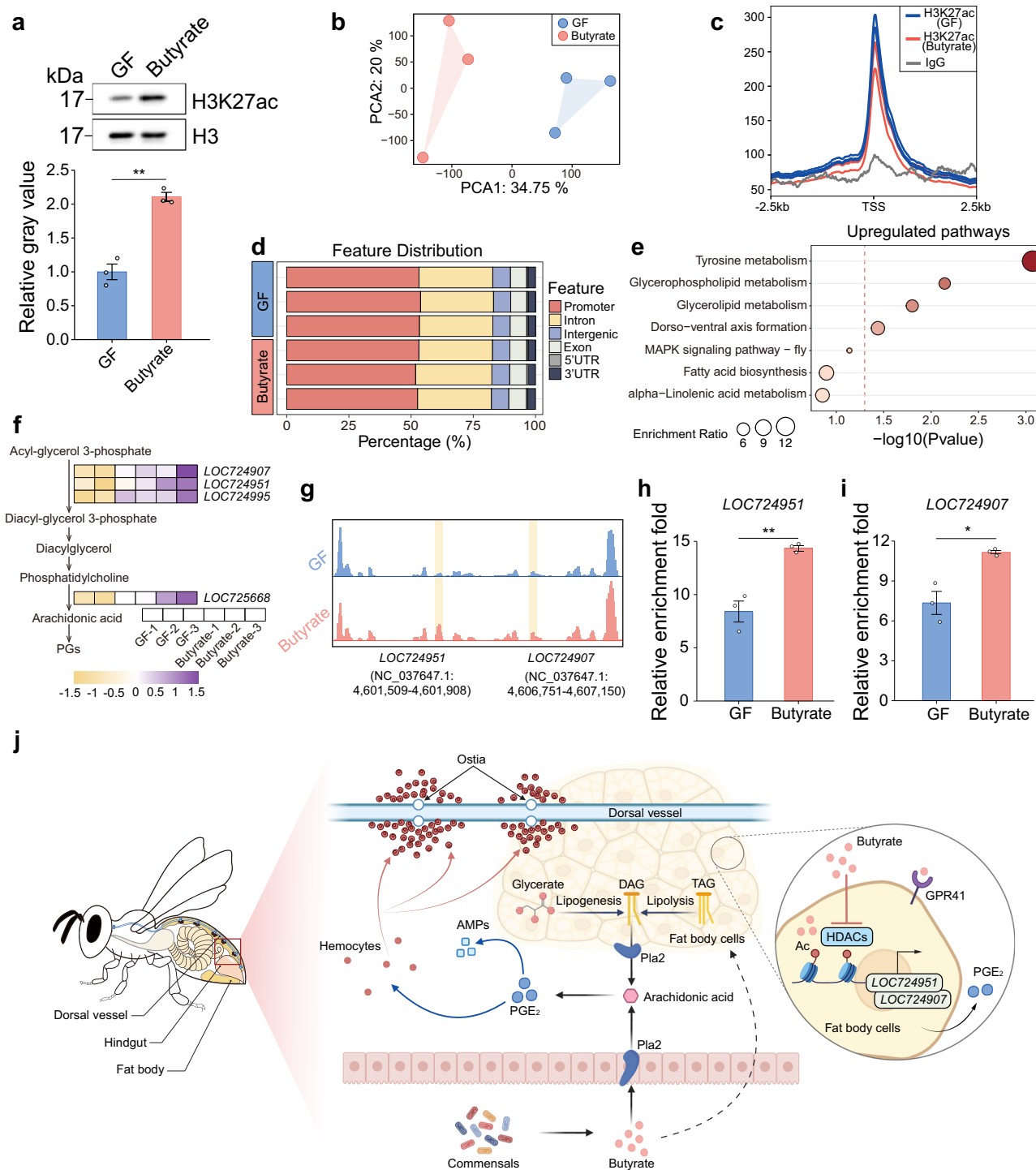

**Fig. 6 | Butyrate regulates lipid metabolism through H3K27ac modification in honey bee fat bodies. a** H3K27ac levels in fat bodies of GF and butyrate-supplemented bees, assessed by western blot (total H3 as the loading control). The relative gray values of western blot were measured using Image J. Bars represent mean ± s.e.m. The images presented are from one of the three independent experiments (see Supplementary Fig. 15a for replicate data). Each sample pooled from 5 individuals from 3 cup cages. $P = 0.0011$ from two-sided Student's $t$ test. **b** Principal Component Analysis based on normalized CUT&Tag peak counts from the fat bodies of GF and butyrate-supplemented bees 5 days post-supplementation. **c** Overlapping peaks of H3K27ac in GF (blue) and butyrate-supplemented (red) groups, with IgG (gray) as the negative control. **d** Distribution of H3K27ac peaks across genomic features (promoters, exons, introns, intergenic regions, 5'UTRs,

and 3'UTRs) for each sample group. **e** KEGG analysis of upregulated genes with differential H3K27ac peaks. $P$ values were determined using a one-sided hypergeometric test. **f** Heatmap of differential H3K27ac peaks in glycerophospholipid metabolism genes. **g** Genomic tracks of *LOC724951* and *LOC724907* from H3K27ac CUT&Tag sequencing. **h, i** ChIP-qPCR of H3K27ac at *LOC724951* (**h**) and *LOC724907* (**i**) promoters. Bars represent mean ± s.e.m. $n = 3$ replicates for each group, with each replicate pooled from 5 individuals from 3 cup cages (two independent experiments, see Supplementary Fig. 15b for replicate data). $P = 0.0044$ (**h**) and 0.0122 (**i**) from two-sided Student's $t$ test. **j** Schematic model illustrating how gut bacteria produce butyrate to prime systemic immunity in honey bees. Source data are provided as a Source Data file. Created in BioRender. Liu, L. (2025) https://BioRender.com/3gkn9pc.

in the bean bug system[19]. Furthermore, we showed that the type of bacterial metabolites involved in priming insect host immunity is more diverse than previously recognized. Specifically, we revealed that, unlike in *D. melanogaster*[9,52], acetate fails to elicit systemic immunity in honey bees. Alternatively, butyrate serves as the key metabolite in the host-gut bacteria interaction. In honey bees, butyrate is likely derived from the microbial fermentation of dietary polysaccharides from pollen cell walls that reach the hindgut. This process may occur via direct breakdown or cross-feeding among core species through known biochemical pathways[53], although the precise genetic determinants require further elucidation.

Our findings demonstrate that a defined consortium of core gut bacteria can prime systemic immunity. While *Gilliamella* and *Snodgrassella* are known to enhance survival[15,16], the collective protection we observed may also originate from the contribution of other symbiotic species, and from potential synergies among them. Moreover, microbial metabolites other than short-chain fatty acids might also contribute to these immune-priming effects.

We further demonstrated that butyrate has induced the upregulation of PGE$_2$, which serves as the key mediator of gut microbiota-induced systemic immunity priming in honey bees. Previous studies have shown that bee gut bacteria increase host-derived prostaglandins[38]. Our study provides evidence that bacterial metabolites can induce PGE$_2$ upregulation in the insect host. While the role of prostaglandins in initiating insect immunity is well known[42,54], their role in immune priming has only been reported in mosquitoes[41]. Our finding thus suggests multiple lineages of insects are capable of utilizing the conserved PGE$_2$ pathway to prime host immunity. Future studies that include more distantly related insect models shall elucidate whether the reported mechanism shared between mosquitoes and honey bees reflects an ancestral state of the insects or an evolutionary convergence.

Based on evidence from the present study, we have discovered a gut-host immune regulatory axis for the insects. The immune response in insects is closely linked to lipid metabolism[55]. For instance, immune activation in *Drosophila* induces lipid metabolic reprogramming to support immune function or to alleviate the ROS stress, as an acute response to exotic bacterial infection[56–58]. Taking advantage of the existing insect machinery, symbiotic bacteria have evolved to trigger host lipid metabolism using their own metabolites, thus regulating host immune responses, e.g., acetate produced by *Drosophila* gut microbiota[52] and butyrate in the honey bee system (present study). Here, we further elucidated the molecular mechanism underlying bacterial orchestration of honey bee lipid reprogramming that occurred in the fat body. Strikingly, the mechanism involved in honey bees is drastically different from what was known from the flies. While *Drosophila* employs an acetate-IMD-neuropeptide Tachykinin-PKA/SREBP axis to suppress midgut lipogenesis[52,59], our results suggested that honey bees utilize butyrate to upregulate TAG catabolic genes, promoting glycerate-to-DAG conversion to enhance PGE$_2$ synthesis. Thus, this gut-host regulatory mechanism unveils a gut microbiota-immune-lipid metabolic axis for the insects. Future studies could directly track lipid metabolic fluxes in vivo, thus testing the model suggested by our transcriptomic data.

Our study also provides in-depth insights into the molecular mechanisms by which butyrate regulates gene expression in honey bees. Specifically, we revealed that SCFAs function through G protein-coupled receptors (GPRs) and epigenetics. Studies based on vertebrates have indicated that SCFAs primarily modulate gene expression via the GPRs or histone deacetylase (HDACs), with GPR41, GPR43, and GPR109 identified as the key molecules[60]. Our findings indicate that the same mechanism (specifically, butyrate via GPR41) is also present in invertebrates. We further showed that this interaction is also achieved through epigenetics. Studies in mammals suggest that butyrate primarily inhibits Class I and Class IIa HDACs[61]. Although

homologous classes are also present in honey bees, isoform-specific antibodies or assays are not yet available to precisely determine which HDAC isoforms are targeted by butyrate in honey bees. Despite the previous recognition that butyrate supplementation increased the overall histone acetylation in honey bees, the specific target genes remain unknown[35]. Here, we employed CUT&Tag to systematically identify H3K27ac-regulated genes under butyrate modulation in invertebrates. To establish causality, a follow-up study could employ RNAi-mediated knockdown in combination with butyrate treatment, which would help determine the specific butyrate-regulated genes. We address that our findings do not necessarily exclude the involvement of alternative epigenetic mechanisms. For example, in addition to HDAC-mediated histone acetylation, butyrate can also mediate non-canonical histone modifications by forming butyrylation marks, such as H3K18bu and H4K12bu, as found in human cells and mouse intestines[62]. These modifications, distinct from acetylation, are capable of enhancing chromatin accessibility and upregulating gene expression[62] and may potentially play a role in honey bees.

Strikingly, our findings indicate that the honey bees share a very similar host-gut interaction mechanism with the mammals, where butyrate generally promotes TAG catabolism[44], induces PGE$_2$ synthesis[63–66], and regulates systemic immune responses[67]. In the context of achieving a shared benefit between parties of an intimate symbiotic system, such an observation might not appear to be surprising. However, it is important to note that the immunity priming mechanisms shared between honey bees and mammals have evolved from completely distinct bacterial communities containing drastically different butyrate-producing bacterial species and sets of host genes underlying PGE$_2$ synthesis and TAG degradation. In mammals, upregulation of PGE$_2$ is associated with increased expression of cyclooxygenase-2 and prostaglandin E synthase[64,66,68,69]. Meanwhile, butyrate enhances TAG catabolism by downregulating peroxisome proliferator-activated receptor γ (PPARγ), thereby suppressing lipid synthesis while promoting fatty acid oxidation[70]. Together with our results, these findings suggest evolutionary convergence in the formation of gut microbiota-host immunity interaction across distinct lineages of life.

In conclusion, our study reveals how gut microbiota regulate the interplay between lipid metabolism and the immune system in honey bees, ultimately improving host survival under pathogen infection. Both honey bees and mammals employ the same bacterial metabolite butyrate to stimulate PGE$_2$ production for immune priming. These findings not only highlight potential pathways for enhancing honey bee immunity in the agricultural and ecological contexts but also advance our understanding of gut microbiota-host interactions from an evolutionary perspective.

## Methods

### Inoculation of germ-free honey bees with conventional, conventional-like, heat-killed gut bacteria, and supplementation of SCFAs

The honey bees used in this study were obtained from an apiary in Changping, Beijing, China (40.1°N, 116.1°E). Within each experiment, bees for all treatment and control groups were derived from a single brood comb to control for genetic background. For each independent replicate experiment, a new brood comb was randomly selected from the apiary containing more than 50 colonies. All experiments were conducted on worker honey bees (*Apis mellifera*), which are female adults.

The GF (germ-free), HK (heat-killed bacteria inoculation), CV (conventional bacteria inoculation), and CL (conventional-like bacteria inoculation) bees were produced as previously described[26]. Briefly, dark-eyed pupae of *Apis mellifera* were removed from capped brood cells using sterile tweezers and reared in sterile plastic cups under controlled conditions (35 °C, 80% humidity) until eclosion. Twenty newly emerged bees under 24-h age were randomly placed into each

sterile plastic cup for bacterial inoculation. The sterility in GF bees was confirmed by absolute quantitative PCR with universal 16s rRNA primers as previously described[27,38].

In the GF group, one milliliter of PBS was mixed with 1 mL of sterile sucrose syrup (50% w/v; Cat#S8271, Solarbio, China) and pollen and provided to the bees. In the CV group, hindguts from 10 nurse bees from their original hives were homogenized with PBS to an $OD_{600}$ of -1, then mixed with 1 mL of sterile sucrose syrup (50% w/v) and pollen before being provided to the bees. In the CL group, bacterial suspensions of five representative gut bacterial strains (Supplementary Table 1) were mixed in equal proportions to a final $OD_{600}$ of -1. This suspension was then mixed with 1 mL of sterile sucrose syrup (50% w/v) and pollen to feed the bees. In the HK group, hindgut homogenates of nurse bees were adjusted to an $OD_{600}$ of -1 with PBS, followed by heat killing at 100 °C for 40 min. This suspension was then mixed with 1 mL of sterile sucrose syrup (50% w/v) and pollen to feed the bees. After being fed with the different mixtures for one day, all groups were provided with sterile sucrose syrup (50% w/v) and pollen under controlled conditions (35 °C, 80% humidity) until the fifth day.

To supplement SCFA, sodium acetate (Cat#A1070, Solarbio, China) or sodium butyrate (Cat#S9491, Solarbio, China) were first dissolved in sterile water to prepare stock solutions, which were then filtered through a 0.22 μm membrane filter (Millipore Cat# SLGP033RB, Thermo Fisher Scientific, USA). The stock solutions were added to the sucrose syrup to a final concentration of 10 mM when feeding the GF honey bees. Newly emerged bees were randomly assigned to groups within 24 h of emergence, and feeding lasted for 5 days. The control group received sucrose syrup containing an equal volume of PBS. Freshly prepared SCFA-containing sucrose solution was provided daily to maintain consistent intake, and consumption was monitored visually by observing the decrease in liquid level in each feeding tube.

For adjusting bacterial cultures to $OD_{600} = 1$, bacterial cultures were washed and diluted with PBS, and 200 μL of the suspension was transferred to a Corning Costar® 96-well cell culture plate. Optical density at 600 nm ($OD_{600}$) was measured using a Molecular Devices i3x multifunctional microplate reader. The PBS dilution was adjusted as needed to achieve an $OD_{600}$ of 1. $OD_{600}$ measurements were performed in triplicate to ensure accuracy.

### Bacteria culture
*Hafnia alvei* was cultured in Luria-Bertani (LB) medium at 35 °C for 16 hr. GFP-expressing *H. alvei* was grown in LB medium supplemented with 100 μg/mL kanamycin at 35 °C for 16 hr. For core gut bacteria, *Gilliamella apicola* W8127 and *Snodgrassella alvi* H11 were cultured for 16 h in Brain Heart Infusion (BHI) medium at 35 °C in a $CO_2$ incubator, while *Bifidobacterium asteroides* W8118 was cultured for 16 h in TPY agar medium under the same conditions. *Bombilactobacillus* W8086 and *Lactobacillus* W8171 were cultured in MRS agar medium at 34 °C in a $CO_2$ incubator for 16 hr. Bacterial strains were revived by streaking onto agar medium. Single colonies were picked, enriched, and identified by 16S rRNA PCR amplification with universal primers, followed by Sanger sequencing and BLAST analysis against the NCBI database to confirm the bacterial species. After cultivation, all bacterial suspensions were washed and diluted with PBS to adjust the optical density at 600 nm ($OD_{600}$) to 1 prior to use.

### Injection procedures
All injection experiments were performed using 10 μL microliter syringes (Shanghai Bolige Industry & Trade Co., Ltd.) fitted with needles of 0.26 mm in diameter. The injected volumes varied depending on the experiment and are specified in the respective Methods sections. In all cases, injections were performed manually through the central dorsal region of the thorax into the hemolymph of honey bees. The solution was delivered slowly over approximately 5 seconds to minimize tissue damage.

### Acetylsalicylic Acid (ASA) and prostaglandin E$_2$ (PGE$_2$) supplementations
The methods for ASA and PGE$_2$ supplementation followed the protocols described in Barletta et al.[40,41]. ASA inhibits heme peroxidase and impairs prostaglandin synthesis in insects[71]. To evaluate the potential toxicity of ASA, CL workers were fed sterile sucrose solution containing 10, 30, or 50 mM ASA (Cat#53526ES03, Yeasen, China) starting on day 3. Supplementation was maintained for 7 days, and survival was recorded daily. Control bees received sucrose solution with an equivalent volume of PBS. Our results showed that 10 mM ASA supplement showed no impact on bee survival and this concentration was thus selected for subsequent experiments. CL bees (day 3) were fed 10 mM ASA for two days and sampled on day 5 for survival assays after *Hafnia* injection or RNA extraction and qPCR analysis.

For PGE$_2$ supplementation, the Ringer solution was prepared according to the protocol described in Costa et al.[72]. PGE$_2$ (Cat#60810ES03, Yeasen, China) was first dissolved in absolute ethanol to prepare a stock solution at 6 μg/μL. The stock was then diluted with Ringer solution at a ratio of 1:11 (stock:buffer), yielding a final working concentration of 0.5 μg/μL. For control treatments, ethanol was diluted with Ringer solution at the same ratio to match the solvent content. On day 5, GF bees were anesthetized by placing the cups containing the bees on ice for 10 min, until the bees were immobilized. They were then injected with 1 μL of the PGE$_2$ solution (0.5 μg) into the hemolymph through the central dorsal region of the thorax using a microsyringe. Control bees were similarly anesthetized and injected with 1 μL of a vehicle-only solution (ethanol diluted in Ringer solution, 1:5, v:v). Three hour post-injection, bees were either dissected to collect abdominal carcasses for qPCR analysis or subjected to hemocyte staining, with dorsal abdomens dissected for confocal imaging. In pathogen infection experiments, bees were injected with GFP-expressing *Hafnia* three hour after PGE$_2$ or vehicle supplementation. Survival rates were monitored, or samples were collected three hour post-infection for immune response analysis.

### Honey bee infection and survival assays
To observe the aggregation of hemocytes in honey bees following systemic infection, we designed a GFP-expressing *Hafnia* mutant[73]. The donor strain *Escherichia coli* MFDpir (carrying a shuttle plasmid with KanR and GFP sequences) was mixed with the recipient *H. alvei* strain ZL01 at a 1:10 ratio. The mixture was incubated for 16 h on Luria-Bertani (LB) medium supplemented with 2,6-diaminopimelic acid (DAP) at a final concentration of 0.3 mM, which is required for the growth of MFDpir. The mixture was then plated on a selective medium containing kanamycin to isolate single colonies expressing GFP. Colony verification was performed using Sanger sequencing with 16S universal primers.

GFP-expressing *Hafnia* were cultured in Luria-Bertani (LB) medium at 37 °C, and adjusted to an $OD_{600}$ of -1 in PBS before injection. Five days after inoculation, GF, HK, CV, CL, and SCFA-supplemented bees were briefly anesthetized on ice for 10 min and injected with 1 μL of GFP-expressing *Hafnia* into the hemolymph using a microsyringe. Following infection, bees were placed back in their respective cups, and mortality was recorded daily for 7 days, with dead bees removed each day. Survival rates were calculated using pooled data from three independent trials.

### Tissue collection
To collect the fat body, we used sterilized tweezers to excise the sting from the terminal abdominal segment and to remove venom gland, gut, ovaries and surrounding tissues, leaving the abdominal carcass intact. The hindgut was also removed and stored at −80 °C for further analysis. Since the fat body is internally lined along the abdominal carcass[74], the remaining abdominal carcass was then used to assess changes to the fat body and stored at −80 °C for gene expression

analysis. Hemolymph was collected from the junction of the head and thorax of the bee using a 10 μL pipettor (Eppendorf, Germany) to measure prostaglandin levels. Approximately 3–5 μL of hemolymph was obtained from each bee, which were maintained in Eppendorf tubes on ice during collection and then stored at −80 °C before analysis. Tissue homogenization was performed using a 1 mL glass homogenizer (Cat#YA0850, Solarbio, China).

## RNA extraction and quantitative PCR

The total RNA of the abdominal carcass or hindgut from each honey bee was extracted using the RNA isolater Total RNA Extraction Reagent (Cat#R401-01, Vazyme, China) according to the manufacturer's instructions. cDNA was then synthesized from 0.5 μg of RNA using the HiScript III RT SuperMix for qPCR (Cat#R323-01, Vazyme, China) in a 20 μL reaction system. The cDNA samples were subsequently diluted 5-fold with nuclease-free water and used as templates for qPCR. qPCR was performed using the Taq Pro Universal SYBR qPCR Master Mix (Cat#Q712-02, Vazyme, China) on a QuantStudio 1 real-time PCR system (Thermo Fisher Scientific, USA) in a standard 96-well block. Each 10 μL qPCR reaction comprised of 1 μL of cDNA template, 0.2 μM of each primer (Supplementary Table 2), and 5 μL of Taq Pro Universal SYBR qPCR Master Mix (Cat#Q712-02, Vazyme, China). The cycling conditions were set as follows: incubation at 95 °C for 3 min; 40 cycles of denaturation at 95 °C for 10 s, annealing/extension at 60 °C for 30 s. The $2^{-\Delta\Delta CT}$ method[75] was employed to normalize the relative gene expression to that of the *actin* gene. All qPCR data are provided in Supplementary Dataset 4.

## Gene interference using dsRNA

The RNAi experiment was designed to investigate the role of the *Pla2* gene in PGE$_2$ production. The dsRNA targeting *Pla2* and control dsRNA targeting GFP were synthesized following the protocol described in Ji et al.[76]. Partial cDNA fragments were cloned into the pCE2-TA-Blunt-Zero vector (Cat#601-01, Vazyme, China), with primers listed in Supplementary Table 2. The dsRNA was synthesized using the T7 Ribo-MAX™ Express RNAi System Kit (Cat#P1700, Promega, USA) and T7 promoter-containing primers. The synthesized dsRNA was mixed with star polycation nanocarriers in a 1:1 mass ratio to prevent degradation and improve delivery efficiency[77]. Each GF bee and butyrate-supplemented bee received an injection of 10 μg of dsRNA into the hemolymph on day 4. The dose refers to the dsRNA content only, excluding the nanocarrier mass. After 24 h, the abdominal carcasses were dissected and used for PGE$_2$ measurements.

## Hemocyte live staining and imaging

At 3, 6, 12, and 24 h after *Hafnia* infection, bees were injected thoracically with 2 μL of a solution containing 67 μM Celltracker CM-DiI (Cat#40718ES50, Yeasen, China) and 1.08 mM Hoechst 33342 (Cat#C0031, Solarbio, China) in PBS. Celltracker CM-DiI specifically labels circulating and sessile hemocytes, while Hoechst 33342 labels all cell nuclei[33]. The Celltracker CM-DiI and Hoechst 33342 mixture was freshly prepared for an optimal hemocyte staining efficiency. Following a 20-min incubation of the staining solution in live bees at 35 °C, five microliters of 8% formaldehyde (Cat#P1112, Solarbio, China) was injected and left to fix for 10 min. Then, the head and thorax of the bee were separated from the abdomen using dissection scissors. The abdomen was split along the coronal plane and placed in PBS containing 0.1% Triton X-100 (Cat#T8200, Solarbio, China) to remove internal tissues. The dorsal abdomen (including the heart) was briefly washed in PBS and mounted between a glass slide and coverslip with an antifade mounting medium (Cat#S2100, Solarbio, China).

Images were acquired at the second abdominal segment, focusing on the dorsal vessel region using a Leica SP8 confocal microscope (Leica Microsystems, Germany) with a 488-nm laser for GFP-expressing *Hafnia*, a 552-nm laser for Celltracker, and a 405-nm laser

for Hoechst 33342. For image analysis, red fluorescence channel images (CellTracker labeling at 552 nm) were imported into ImageJ v1.53a (NIH, Bethesda, USA), converted to 8-bit, and inverted. A threshold was applied based on the actual distribution of labeled hemocytes to select positive signals while excluding background fluorescence. The threshold was set according to consistent criteria across all samples to ensure reproducibility. The percentage of positive fluorescent area relative to the total image area was calculated as a measure of hemocyte number. Since this metric is based on relative fluorescent area rather than absolute intensity, the results are directly comparable across samples.

## Nile Red staining and triacylglycerol (TAG) measurements

The abdominal carcass was dissected in PBS, fixed with 4% formaldehyde for 30 min, washed three times with PBS, stained with Nile Red (5 μg/mL) and DAPI (10 μg/mL) for 40 min. After staining, it was washed three times with PBS and mounted using an antifade mounting medium. Confocal images were captured using a Leica SP8 confocal microscope. For TAG detection, the abdominal carcasses of three bees were homogenized in 500 μL of PBS. The samples were centrifuged at $3000 \times g$ for 10 min at 4 °C, and 10 μL of the supernatant was used for TAG measurement using an insect TAG ELISA kit (Cat#TW1782H, Shanghai Tongwei, China).

## Measurement of butyrate concentrations using LC-MS

Three bee abdomens were pooled per biological replicate, minced using dissection scissors, and transferred to a microcentrifuge tube. Deionized water was added to a final volume of 2 mL. The sample was homogenized and underwent ultrasonic decomposition for 10 min. Following centrifugation at $15,000 \times g$ at 4 °C for 10 min, the supernatant was filtered with a 0.22-μm filter (Corning, USA) for LC-MS analysis. Each treatment group consisted of three biological replicates. A Thermo Scientific Dionex ICS-5000+ ion chromatography system (Thermo Fisher Scientific, USA) was used to analyze butyrate concentrations. The system was equipped with an SP single pump (ICS-5000), an EG eluent generator (ICS-5000), an anion suppressor (AERS 4 mm), an AS-AP auto-sampler, a DC detector (conductivity and electrochemical detectors), and Chromeleon 7.2 SR5 chromatography data system. Chromatographic separation was performed using a Dionex IonPAC AS11-HC analytical column (4×250 mm) with a Dionex IonPAC AG11-HC guard column (4×50 mm). The mobile phase, KOH, was automatically generated by an eluent generator (EGC 500 KOH) containing 1 L of 0.5 M high-purity KOH solution. The analysis was carried out with a running time of 28 min at a flow rate of 1 mL/min and a system pressure of approximately 1,700 psi. The suppressor (AERS-4 mm) operated at a suppression current of 12 mA. The detection cell heater temperature was set to 35 °C, with the column compartment maintained at 30 °C and the sample chamber at 8 °C. Ultrapure water was used to ensure the quality and consistency of the mobile phase. A 10 μL volume of the supernatant solution was injected for analysis.

## Prostaglandin E$_2$ (PGE$_2$) ELISA assay

The whole abdomen, hindgut, and hemolymph of GF, CV, CL, and butyrate-supplemented bees (day 5) were collected for PGE$_2$ analysis. For tissue samples, five entire abdomens or hindguts were pooled as one biological replicate. The tissues were minced, homogenized in 1 mL of pre-chilled PBS containing protease inhibitors on ice, and centrifuged at $5000 \times g$ for 15 min at 4 °C. The supernatant was collected and stored at −80 °C until further analysis. For hemolymph samples, at least 25 μL of hemolymph was extracted from five bees using a 10 μL pipettor and transferred to a centrifuge tube. The hemolymph samples were then stored at −80 °C until further analysis. In the *Pla2*-RNAi experiments, abdominal carcasses were collected 24 h after RNAi treatment from GF bees treated with GFP dsRNA, butyrate-supplemented bees treated with GFP dsRNA, and butyrate-

supplemented bees treated with *Pla2* dsRNA. Five abdominal carcasses were pooled per biological replicate, processed as described above, and stored at −80 °C for subsequent analysis.

The PGE$_2$ levels were measured using a PGE$_2$ ELISA kit (Cat#D751014, Sangon Biotech, China) according to the manufacturer's protocol. Briefly, 50 μL of standards or samples were added to each well, followed immediately by 50 μL of biotin-labeled PGE$_2$ antibody working solution. After incubating at 37 °C for 45 min, the wells were washed five times with 300 μL of washing buffer. Subsequently, 100 μL of HRP-labeled streptavidin solution was added, and the plate was incubated at 37 °C for 30 min. After another washing step, 90 μL of substrate solution was added, and the plate was incubated in the dark at 37 °C for 15 min. Finally, 50 μL of stop solution was added, and OD values were measured at 450 nm using Spectramax i3x (Molecular Devices, USA). PGE$_2$ concentrations were calculated using a standard curve based on a four-parameter logistic model.

### Trichostatin A (TSA), AR420626, β-hydroxybutyric acid (SHB) Supplementation and HDAC activity assay

To evaluate whether butyrate exerts its effects via HDAC inhibition, GF bees (day 4) were injected with 2 μL of TSA (Cat#58880-19-6, Cayman, USA) solution (0.1 mM), a classic HDAC inhibitor. The TSA solution was prepared by dissolving TSA in absolute ethanol and diluting it with Ringer solution. Control bees were injected with 2 μL of a mixture of ethanol and Ringer solution. Butyrate-supplemented bees were treated in the same way as control group. After 24 h, abdominal carcasses were dissected for HDAC activity analysis, qPCR or Nile Red staining. To assess HDAC activity, five abdominal carcasses were pooled to form one biological replicate. Tissues were dissected and homogenized in 1 mL of ice-cold PBS. After centrifugation at 3000 × *g* for 10 min at 4 °C, the supernatant was collected and analyzed using an insect HDAC ELISA kit (Cat#TW1695H, Shanghai Tongwei, China).

To investigate whether butyrate functions through GPCR signaling pathway, honey bees were treated with either a GPR41 antagonist, β-hydroxybutyric acid (SHB; Cat#300-85-6, MedChemExpress, USA), or a GPR41-specific agonist AR420626 (Cat#1798310-55-0, Cayman, USA).

For SHB treatment, SHB powder was dissolved in Ringer solution to prepare a 400 mM stock solution, which was then filtered through a 0.22 μm membrane filter (Millipore Cat#SLGP033RB, Thermo Fisher Scientific, USA). To determine the appropriate supplementation concentration, SHB was added to 10 mM sodium butyrate-sucrose solution to final concentrations of 20 mM, 50 mM, and 100 mM. GF and butyrate-supplemented bees received sucrose solution containing equivalent volumes of Ringer buffer. Newly emerged bees (<24 h) were randomly assigned to groups and fed with the respective solutions for 7 days, during which survival was recorded daily. Based on the survival results, 20 mM and 50 mM SHB were selected for subsequent supplementation experiments.

For qPCR experiments involving SHB supplementation, GF and butyrate groups received sucrose solution supplemented with equivalent volumes of Ringer buffer. Newly emerged bees ( < 24 h) were randomly assigned to groups and fed with the respective solutions for 5 days. On day 5, abdominal carcasses were dissected for qPCR analysis.

For AR420626 treatment, the AR420626 solution was prepared in the same manner as TSA and diluted to a final concentration of 1 mM. GF bees (day 4) were injected with 2 μL of the AR420626 solution, while control bees received an equivalent volume of a mixture containing ethanol and Ringer solution. After 24 h, abdominal carcasses were collected for qPCR or Nile Red staining.

### Protein extraction and western blot analysis

The abdominal carcasses of GF bees and butyrate-supplemented bees (day 5) were dissected to extract histones for western blot analysis.

Histone extraction was performed using an Insect histone extraction kit (Cat#EX1540, Solarbio, China) following the manufacturer's protocol. Briefly, five abdominal carcasses were pooled as one biological replicate. The dissected abdominal carcasses were homogenized in 500 μL of lysis buffer containing protease inhibitors, centrifuged at 16,000 × *g* at 4 °C for 15 min, and the pellet was resuspended in histone extraction buffer. After incubation overnight at 4 °C, the mixture was centrifuged at 16,000 × *g* at 4 °C for 10 min to collect histones in the supernatant.

Histone concentration was quantified using a BCA protein assay kit (Cat#P0012S, Beyotime, China), and 50 μg of protein per sample was loaded onto a 5–12% SDS-PAGE gel for electrophoretic separation. Proteins were then transferred onto a PVDF membrane, which was blocked with 5% non-fat milk and incubated with primary antibodies overnight at 4 °C. The membrane was subsequently incubated with secondary antibodies (1:2000 dilution) at room temperature for 1 h. Protein blots were detected using the Amersham ImageQuant 800 imaging system (GE Healthcare). The primary antibodies used were Acetyl-Histone H3 (Lys27) (D5E4) XP® Rabbit mAb (Cat#8173, Cell Signaling Technology, USA) and Anti-Histone H3 Mouse Monoclonal Antibody (Cat#BE3015, Easybio, China). The secondary antibodies used were HRP-conjugated Goat Anti-Rabbit IgG (H + L) (Cat#SA00001-2, Proteintech, China) and HRP-conjugated Goat Anti-Mouse IgG (H + L) (Cat#SA00001-1, Proteintech, China). Signal intensity was quantified using ImageJ software, and histone modification levels were normalized to histone H3 signals. Uncropped and unprocessed full scan blots are provided in the Source Data file, and those for Supplementary Figs. are included in the Supplementary Information.

### Transcriptome sequencing and analysis

Total RNA was extracted from the abdominal carcass or hindgut of GF bees and those supplemented with butyrate for 5 days. Library preparation and transcriptome sequencing were performed by Novogene (Novogene, China). RNA quality was determined using an Agilent 2100 Bioanalyzer (Agilent Technologies, USA) and agarose gel electrophoresis. Sequencing libraries were constructed using the NEBNext Ultra RNA Library Prep Kit for Illumina (NEB, USA), and PCR products were purified with the AMPure XP system. The library quality was analyzed using the Agilent 5400 system (Agilent, USA), and sequencing was conducted on the Illumina NovaSeq platform, producing 150 bp paired-end reads. Each group included 3 biological replicates, with each replicate comprising the abdominal carcass or hindgut of a single honey bee.

Raw reads with a quality score below 20 in over 10% of bases were discarded using fastp (version 0.13.1)[78]. Clean reads were mapped to the *Apis mellifera* reference genome (Amel_HAv3.1) using HISAT2 (version 2.1.0)[79], and the aligned reads for each sample were assembled using StringTie (version 2.0.6)[80]. Differential gene expression analysis was performed using the DESeq2 package (version 1.38.3)[81] in R (version 4.2.3), with significantly differentially expressed genes (DEGs) defined by thresholds of an adjusted p-value < 0.05 and |log2-fold change| ≥ 1. Pathway analysis of DEGs was conducted with the R package ClusterProfiler (version 4.6.2)[82] based on KEGG Orthologue annotations.

### Untargeted metabolomic analyses

The abdomens of GF bees and bees supplemented with butyrate for 5 days were dissected and rapidly frozen in liquid nitrogen. Metabolomic sequencing was performed at Majorbio Biotech (Majorbio Biotech, China) using six biological replicates per group. A sample of 50 mg was weighed into a 2 mL centrifuge tube with a 6 mm grinding bead. 400 μL of extraction solution (methanol:water = 4:1, v/v) containing four internal standards (e.g., L-2-chlorophenylalanine at 0.02 mg/mL) was added. The mixture was ground for 6 min at −10 °C (50 Hz) and subjected to ultrasonic extraction for 30 min at 5 °C

(40 KHz). After standing at −20 °C for 30 min, the sample was centrifuged at 13,000 × g for 15 min at 4 °C. The supernatant was transferred into a sample vial for analysis, and 20 μL from each sample was pooled for quality control. The LC-MS analysis was performed on a Thermo Fisher UHPLC-Q Exactive HF-X system (Thermo Fisher Scientific, USA) equipped with an ACQUITY UPLC HSS T3 column (100 mm × 2.1 mm i.d., 1.8 μm; Waters, Milford, USA). Chromatographic separation was achieved using gradient elution with 95% water +5% acetonitrile containing 0.1% formic acid (solvent A) and 47.5% acetonitrile +47.5% isopropanol +5% water containing 0.1% formic acid (solvent B). The injection volume was 3 μL, and the column temperature was maintained at 40 °C. Mass spectrometry detection utilized an electrospray ionization source operated in both positive and negative ion scanning modes. The spray voltage was set to 3.5 kV for the positive mode and −3.5 kV for the negative mode. Sheath gas and auxiliary gas flow rates were adjusted to 50 and 13 arbitrary units, respectively, with the capillary temperature maintained at 325 °C. Full MS scans were acquired at a mass range of $m/z$ 70–1050 at a resolution of 60,000. Normalized collision energies of 20, 40, and 60 eV were applied during MS/MS analysis.

LC/MS raw data were processed using Progenesis QI software (Waters Corporation, USA) for baseline correction, peak detection, integration, retention time alignment, and peak normalization, resulting in a data matrix containing retention time, mass-to-charge ratio ($m/z$), and peak intensity information. Metabolite identification was performed by searching against databases with a mass error threshold of less than 10 ppm, using HMDB, Metlin, and the Majorbio database as primary references. Data analysis was performed on the Majorbio Cloud Platform (www.majorbio.com). Metabolites with more than 20% missing values within any group were excluded, while missing values caused by low signal intensities were imputed with the minimum value. Mass spectrum peak response intensity was normalized using the sum normalization method, followed by $\log_{10}$ transformation to reduce errors from sample preparation, concentration differences, and instrument variability. Variables with a relative standard deviation (RSD) ≤ 30% in QC samples were retained. Statistically significant differences between the GF and butyrate supplemented groups were identified with a threshold of Variable Importance in Projection (VIP ≥ 1), |Fold change | ≥ 1, and P ≤ 0.05 by Student's $t$ test analysis. KEGG pathway enrichment analysis for differential metabolites was performed based on the hypergeometric distribution algorithm. The $p$-values were adjusted using the BH method, and pathways with corrected $p$-values < 0.05 were considered significantly enriched.

### Nuclei extraction, CUT&Tag library construction, sequencing and analysis

Nuclei were isolated from abdominal carcasses using a Nuclear extraction kit (Cat#52009-10, Shbio, China) following the manufacturer's protocol for subsequent CUT&Tag library construction. Abdominal carcasses from GF and butyrate-supplemented bees (day 5) were dissected, with five carcasses pooled as one biological replicate. The abdominal carcasses were homogenized in lysis buffer. After 10 min of lysis, the homogenate was filtered through a 40 μm cell strainer (Cat#431750, Corning, USA) to remove debris and transferred to a centrifuge tube, and centrifuged at 500 × g at 4 °C for 5 min. The pellet was resuspended in LB, PB1, PB2, and PB3 solutions provided in the kit to induce phase separation. The nuclear layer was extracted and filtered again through a 40 μm cell strainer. The extracted nuclei were stained with Trypan blue (Cat#C0040, Solarbio, China) to assess the quantity and integrity of the extracted nuclei.

The CUT&Tag library was prepared using the Hyperactive Universal CUT&Tag Assay Kit for Illumina Pro (Cat#TD904, Vazyme, China) following the manufacturer's protocol. Nuclei were bound to Concanavalin A-coated Magnetic Beads Pro, and incubated overnight with the primary antibody at 4 °C. The nuclei-bead complexes were then incubated with a diluted secondary antibody (1:100) at room temperature for 1 h. The pA/G-Tnp Pro transposase was added to the nuclei-bead complexes and incubated at room temperature for 1 h to cleave DNA sequences near target proteins. DNA was extracted using DNA Extract Beads Pro, and the libraries were amplified using Eppendorf AG 22331 Hamburg PCR Thermal Cycler. The PCR products were purified with VAHTS DNA Clean Beads (Cat#N411-01, Vazyme, China) for library construction. The primary antibodies used were Acetyl-Histone H3 (Lys27) (D5E4) XP® Rabbit mAb (Cat#8173, Cell Signaling Technology) and Rabbit IgG (Cat#A7016, Beyotime) as a negative control. The secondary antibody used was Goat Anti-Rabbit IgG H&L (Cat#Ab207-01, Vazyme, China).

DNA library quality assessment and sequencing were conducted by Novogene. The library concentration was initially measured using Qubit, and the fragment integrity was assessed using AATI. The effective library concentration was determined by qPCR. Sequencing was performed on the Illumina NovaSeq platform with paired-end 150 bp reads (PE150). CUT&Tag sequencing was conducted using three biological replicates. Raw reads were processed with Trim Galore (version 0.6.10)[83] and aligned to the *Apis mellifera* reference genome (Amel_HAv3.1) using Bowtie 2 (version 2.3.4.3)[84]. Duplicated and unmapped reads were removed using Sambamba (version 0.6.6)[85] and Samtools (version 1.9)[86]. Enriched peaks were called with MACS2 (version 2.2.6)[87]. Peak annotation was conducted with R package ChIPseeker (version 1.43.0)[82]. Differential peak analysis was performed with the R package DiffBind (version 3.8.4)[88], with significantly differential peaks defined as those with p-value < 0.05 and Fold change > 1.5[89]. KEGG enrichment analysis was performed using the R package ClusterProfiler (version 4.6.2)[82]. ChIP-qPCR was performed as described previously[90]. The DNA was treated with Stop Buffer (Cat#TD904-C1, Vazyme, China) and quantified by real-time PCR using the primers listed in Supplementary Table 2. The DNA Spike-in was used as an internal control to calculate ΔCT values, and fold enrichment relative to IgG was determined using the $2^{-\Delta\Delta CT}$ method[75].

### Statistics and reproducibility

Statistical analyses were performed in GraphPad Prism (v9.3.1).

Comparisons between two groups used unpaired Student's $t$ tests when data met assumptions of normality and homogeneity of variance, Welch's $t$ tests when variances were unequal, and Mann–Whitney tests for non-normal distributions. For multiple comparisons, one-way ANOVA with Tukey's HSD test was applied when the assumptions of normality and homogeneity of variance were satisfied. Brown–Forsythe and Welch ANOVA were used for normal data with unequal variances, and the Kruskal–Wallis tests followed by Dunn's test were applied otherwise. Statistical significance was defined as $P < 0.05$.

No statistical method was used to predetermine sample size. No data were excluded from the analyses. The investigators were not blinded to allocation during experiments and outcome assessment.

### Reporting summary

Further information on research design is available in the Nature Portfolio Reporting Summary linked to this article.

## Data availability

The sequencing data were uploaded to NCBI Bioproject with an accession number PRJNA1232651 and PRJNA1233551. The metabolomic raw data were uploaded to Metabolights with accession number MTBLS13098. Source data are provided with this paper.

## Code availability

The scripts for RNA-seq and CUT&Tag analysis have been deposited on GitHub and archived in Zenodo under https://doi.org/10.5281/zenodo.

18043934, and Figshare [https://doi.org/10.6084/m9.figshare.30403936].

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

## Acknowledgements

The authors thank Prof. Hao Zheng from China Agricultural University for providing honey bee gut bacterial strains used in this study, Prof. Jeffrey Barrick from Michigan State University for donating the MFDpir strain, Prof. Jie Shen and Prof. Shuo Yan from China Agricultural University for providing nanocarriers used in RNAi experiments, Prof. Ken Tan and Prof. Zhengwei Wang from Xishuangbanna Tropical Botanical Garden, Chinese Academy of Sciences for their assistance in honey bee management and sampling, Assistant Prof. Xue Zhang from China Agricultural University, Dr. Lizhen Guo, Dr. Chengfeng Yang, Dr. Qinzhi Su and Dr. Chenyi Li for assistance in *Hafnia* isolation, maintenance and modification, the Public Technology Service Center of Xishuangbanna Tropical Botanical Garden, Chinese Academy of Sciences for providing essential infrastructure and instruments. The work was supported by the Ministry of Science and Technology of the People's Republic of China (2023YFD2201804) and the 2115 Talent Development Program of China Agricultural University to X.Z., and the National Natural Science Foundation of China (32000343) to S.L.

## Author contributions

Conceptualization and supervision: S.L. and X.Z.; Data acquisition: J.L., Z.L., J.T.; Data analysis: J.L., Y.W.; Writing—original draft: J.L., S.L., and X.Z.; Writing—review & editing: J.L., Z.L., Y.W., J.T., X.Z. and S.L.

## Competing interests

The authors declare no competing interests.
