## [Transparent Peer Review file · Nature Communications]

Gut microbiota-derived butyrate primes systemic immunity in honey bees by mediating lipid metabolic reprogramming

Corresponding Author: Dr Shiqi Luo

Version 0:

Reviewer comments:

Reviewer #1

(Remarks to the Author)

This study investigates how the honey bee gut microbiota, particularly the metabolite butyrate, primes systemic immune responses. Using germ-free bees and supplementation experiments, the authors show that butyrate triggers lipid metabolic reprogramming in the fat body. This leads to prostaglandin E2 (PGE2) synthesis and enhanced immune gene expression. The work reveals a novel microbiota–metabolite–immunity axis conserved across insects and mammals.

Impressive multi-omics integration (transcriptomics, metabolomics, CUT&Tag, qPCR); well-designed experimental contrasts (GF, HK, CV, CL); strong mechanistic narrative linking butyrate to PGE2 to immunity.

What stats software was used and where are the codebases for statistical works?

Line 78 and this section: other sections used dot plots and bar charts to show sample sizes (e.g. fig 2) this section did not and I found it tedious to find the sample sizes.

Line 78 onwards: Any indication that these effects are consistent across multiple colonies/genotypes? Context on bee genetic background is missing. How many colonies? Were they MDIs or SDIs? How much colony-level variation explained your data, etc? I worry that with the smaller sample size (e.g. for gene expression) your stats are not necessarily robust.

Lines 61-75: Consider making the last sentence more specific about the downstream applications or broader impact.

Line 132: Would be helpful to mention natural butyrate levels in bees for ecological relevance and to have a citation here.

Lines 189–225: It's unclear if transcript-level differences translate into functional lipid mobilization; could add a metabolic flux analysis.

Lines 321–363: While I find you have a compelling dual mechanism (GPR41 and HDAC inhibition), isn't HDAC activity here only inferred from TSA and H3K27ac? Consider discussing limitations of specificity.

(Remarks on code availability)

The code and data sets are available but the code should be on something more permanent than github. I don't see the data yet, but they have provided accessions.

The code does also look to just be the alignment-based pieces. No statistical pieces have been included here.

Reviewer #2

(Remarks to the Author)

The manuscript titled "Gut microbiota-derived butyrate primes systemic immunity in honeybees by mediating lipid metabolic reprogramming" describes a regulatory mechanism unveiling a novel gut microbiota-immune-lipid metabolic axis for insects. The study is based on immune challenge under different microbiota backgrounds, followed by metabolomics and epigenomics analyses, offering an in-depth characterization of the molecular mechanisms underlying immune priming via gut in honeybees.

The study is accurate, the experimental methods sound and adequate. The results are well presented and provide interesting new information on insect immunity and, more in general on animal immunity. The data interpretation and discussion are adequate and require only some additional considerations as indicated in the comments below.

We have only a few suggestions that the authors may want to consider for revision.

-line 40: Please, check reference 15. This article shows that *Lactobacillus* strains trigger gut immune response, against gut opportunistic pathogens of honeybee. It seems apparently not related to systemic immunity. Instead, you may want to consider articles such as Kwong et al 2017, in which apidaecin expression in hemolymph is studied in response to gut microbiota (DOI: 10.1098/rsos.170003).

-line 83: "gut homogeny" should be "gut homogenate"?

-figure 1C: we suggest analyzing the data using 2-way anova + post-hoc test, using different letters to indicate differences. Moreover, qPCR data should be plotted using GF at time 0 (eg. 3hours) as calibrator, to provide a dynamic picture of the time-course expression. In fact, in the present figure, which showed each time point separately, the AMPs expression curve of GF bees is unrealistically flat (ie. expression of AMPs can't be always 1).

-line 129: "triggers systemic immunity priming" stands for "primes systemic immunity"?

-lines 130-131: "We applied 10 mM of acetate and butyrate to the bees to examine their role in increasing bee immune responses observed in this study". This sentence sounds a bit "dropped" without any reference to previous studies, hypothesis or findings. Why acetate and butyrate? Did you identify those metabolites in bees with functional GM? Or this is just a hypothesis generated by findings in mammal models? A short introductory sentence addressing this issue is required.

-lines 497-499: The discussion section would benefit from few lines about the origin of butyrate. From figure 2b, it seems that all core species produce butyrate, although at different levels. Do they have the same metabolic capabilities to produce this metabolite? May butyrate be metabolized from dietary fibers, such as in other animals?

-line 585: Indicate which model of microsyringe, which needle gauge and where bees were injected (e.g. between which abdominal segments? laterally, dorsally?).

-line 627: Which nanocarriers? Please, add this information

-Data availability: We appreciated the availability of RNAseq raw data and code, but we cannot find qPCR and metabolomic raw data + analysis/code.

(Remarks on code availability)

Code for RNAseq experiments is working, but I cannot find code used for stats and plot of qPCR and metabolomics data.

Reviewer #3

(Remarks to the Author)

The manuscript "Gut microbiota-derived butyrate primes systemic immunity in honeybees by mediating lipid metabolic reprogramming" presents a promising study into how microbiota-derived metabolites, such as butyrate, modulate immune function in honeybees. The concept is timely and compelling, with potential relevance not only to bee studies but also to broader questions of host-microbe interactions and immune education. The authors tackle this through a multi-omics approach: microbiome manipulations using germ-free, heat-killed, conventionally colonized, coupled with targeted metabolite supplementation (e.g., butyrate and acetate) and metabolite quantification via LC-MS. They conducted pathogen challenge assays with the microbe *Hafnia* to assess immune competence and host survival. The study further integrated transcriptome sequencing and untargeted metabolomic analysis, qPCR for gene expression, assessment of lipid metabolism, and imaging of hemocyte aggregation amongst others, making the overall dataset rich and multidimensional. However, while the core idea is strong and the experimental breadth is commendable, several key issues need to be addressed to clarify data interpretation, strengthen the conclusions, and improve reproducibility. Below are comments detailing these concerns.

Major Comments:

- Some of the data interpretation in the manuscript is confusing. A few examples of this include:
 - o Lines 91-94 imply that immune gene expression is significantly different in CV vs HK bees, while Figures 1C and S1B show that the relationship between these conditions are much more complex, and not significantly different across the board as implied.
 - o The acetylsalicylic acid experiment is lacking the proper controls. In particular, there are no data shown for the effect of ASA treatment in a naïve bee. In particular, it needs to be demonstrated that the effect of ASA treatment is specifically decreasing survival post-infection because of the infection, rather than an indiscriminate impact on bee survival following ASA administration.
- The hemocyte area experiments are especially difficult to interpret:
 - o The stained images are incredibly small and dark and don't appear to support the text descriptions. In particular, the DNA stain is very scattered, and doesn't seem to stain any of the proposed hemocytes.
 - o "Hemocyte area" is a meaningless measure without more context (such as measures of cell density and cell size). Many insects have changes in cell number as well as the differentiation of distinct immune cells types following infection. Cell number and identity need to be determined directly.

- o There is no context/description for why hemocyte aggregation would be a meaningful measure of immune competence. Is it a rough proxy for total immune cell numbers? Is this location somehow relevant to immune responses?
- o Hemocytes are stained with CM-Dil. This has been shown to stain a class of hemocytes in mosquitoes, but there is no data or citation presented to validate that this stain is effective in bees. Particularly, it needs to be demonstrated that the CM-Dil signal is specific to hemocytes, and to determine which class(es) of hemocytes are stained by CM-Dil.
 - The experiments to test the role of butyrate are not definitive and do not support the conclusions drawn. The use of AR420626 and TSA do not explicitly test the effect that butyrate is having in bee immunity. They independently act in pathways previously linked to butyrate so are suggestive, but statements such as lines 334-335 "butyrate regulates lipid break-down in honeybees via both GPR41 activation and histone acetylation modulation" are not adequately supported by the data. More specific experiments are necessary to support this definitive conclusion.
- o At a minimum a more realistic and supportable statement of the findings is necessary.
- o More direct experiments would considerably strengthen the conclusions of the manuscript – for instance, treating bees with butyrate + an appropriate GPCR receptor antagonist would allow for a stronger conclusion.
 - The methods are incompletely described, and so it would be difficult to replicate this work. The entire section needs to be improved, please see some specific examples below.
 - The statistical analysis section is lacking in detail. Tests are listed, but in many instances, the test used in a specific experiment is not definitely stated, either in the methods section or the figure legends.
 - The authors should acknowledge the methodological and other limitations of the study for a more thorough discussion.

General/Minor Comments:

- The introduction (e.g. lines 28-31) references immune priming in the context of establishing immune memory. The rest of the manuscript uses a distinct definition of "priming", which is closer to establishing immune competence. This aspect of the introduction should be rewritten to better reflect the findings of the manuscript.
- Lines 48-49 refer to bacteria species by outdated names (Firm-4 and Firm-5) which have been replaced by more descriptive names. Information is available in the Motta and Moran, 2023 review (citation 20).
- Some of the writing is too specialized for publication in a general-interest journal. For example, line 63, refers to the pathogen *Hafnia* without giving a complete species name or any background information. Additionally, the section continues by discussing experimental treatments (lines 65-67) such as conventionally colonized, heat-killed, and germ-free, but without giving sufficient detail for someone outside the field to interpret the context of the experiments.
- The study emphasizes butyrate as a central modulator of immune responses in honeybees. However, Line 68 raises the question of why butyrate was selected as the primary focus over other structurally similar metabolites. The authors briefly mention acetate, but it would strengthen the rationale to elaborate on any prior evidence or biological plausibility supporting butyrate's unique systemic effects in bees.
- Lines 98–100 (Figure 1): Is there a temporal resolution of immune activation? E.g. 24h GF compared to 3hr GF. 24h HK compared to 3h HK and is 12h CV higher than 3h CV? How does time post infection affect the hemocyte aggregation? How does 24h post infection have some level of aggregation (Figure 1F) yet the relative expression shown is very low.
- Line 161-163: There is a discrepancy between current findings and prior reports (e.g., Horak et al., 2020) on the immunogenicity of heat-killed bacteria which deserves discussion. This study states that heat-killed gut bacteria showed no impacts on hemocyte aggregation, with or without pathogen infection, whereas previous work has suggested HK *S. alvi* can trigger immune responses. The authors should discuss their ideas underlying this difference.
- Line 173: Although the CL group shows high post-infection survival, the study does not directly test whether the effect is due to synergistic interactions among the five core microbes. The discussion would benefit from either a mechanistic hypothesis or additional data to explain this enhanced protection.
- Line 330: The study suggests HDAC inhibition by butyrate and TSA but does not specify which HDAC enzymes/isoforms are affected. Given the functional divergence among HDAC classes, this detail could clarify the pathway specificity.
- Line 132 says that acetate and butyrate were chosen "in approximation of natural conditions" but there are no data presented or cited which show the presence or concentration of these metabolites in bee hemolymph.
- The context for the CL treatment group is not fully explained. It appears that in natural conditions, bees don't have equal amounts of the core microbiome species (e.g. Motta and Moran, citation 20). The results seem comparable to the presumably more natural CV treatment, but it was a bit confusing to follow without a full explanation.
- Lines 263-264 make reference to the substrate for prostaglandin. Prostaglandin is a lipid compound rather than an enzyme, so it's unclear what "substrate" implies in this context.
- Line 264 is also confusing ... "RNAi of Pla2 reduced its expression" – what is "its" referring to. RNAi of Pla2 reduces Pla2 expression? Or one of the other molecules discussed in the preceding sentences?
- Confusing wording: "Cohesively" in line 330
- Line 332 refers to expression effects on the "aforementioned genes", but the genes presented are not the same as the genes from the previous result (which I would assume is meant by aforementioned genes). The lists have considerable overlap, but are not identical as implied by the text.

Methods comments :

- Generally, clarify the number of biological replicates per group for each assay and trial. Indicate the number of independent experiments and the number of bees or cups used per group in each trial to ensure reproducibility and statistical robustness.
- Line 558 states that bees were fed ASA on day 3, but doesn't specify the length of exposure.
- Sequence accessions are not given for the genes of interest.
- Line 627 states that dsRNA was mixed with "nanocarriers" without naming/describing the nanocarrier used.
- Line 629: Does the dose of 10 µg refer to RNA alone? Or RNA + nanocarrier?
- Line 650 should describe how the fluorescence images were analyzed. It fails to mention the measure used, how fluorescence intensity differences were normalized/controlled, etc.

- Include methods used to confirm sterility in GF bees. Was sterility verified via plating, PCR, or another microbial contamination check? Also, specify whether any antibiotics other than kanamycin were used for strain maintenance or selection for the other bacteria species.
- Report the duration of bacterial culture (number of hours grown before use). "Overnight" is ambiguous.
- Describe how OD600 was measured (e.g., spectrophotometer model, and cuvette size).
- Indicate how bacterial strain identity and purity were verified—such as by 16S rRNA sequencing, PCR with species-specific primers, or phenotypic methods?
- Specify (briefly) the role and concentration of any additives used (e.g., 2,6-diaminopimelic acid).
- Indicate when and for how long SCFAs were administered, and describe whether consumption was monitored (e.g., measuring syrup intake or visual confirmation).
- Include the preparation steps for PGE2: initial concentration in ethanol, subsequent dilution series (e.g., 1:10 dil) and final working concentration so that the final ethanol content can be assessed
- Clarify the injection site for PGE2 (e.g., thorax vs abdomen, specific tergite location).
- Describe the duration of anesthesia and how it was induced (e.g., ice exposure for 5 minutes).
- Provide injection parameters: volume of injected solution, needle gauge/type, and injection rate if known (e.g., $\mu\text{L}/\text{sec}$).
- Indicate whether any anticoagulants were used during hemolymph collection.
- Identify the type of homogenizer used and the specific lysis buffer (name and composition, if applicable).

Figures:

- General: The figure legends are ambiguous about the statistical tests, and give options for tests which may have been performed rather than directly stating which test led to the significance determinations shown in the figures.
- General: Many of the figure legends don't describe the figure fully. For example, several of the hemocyte stain images have blue signal, but it is not specified in the legend as the red and green signals are.
- Figure 1A: The schematic is difficult to interpret. It implies that bees were sacrificed for gene expression following survival analysis, whereas the text implies these were separate experiments. The schematic also shows that inoculation happened at Day 0, but the legend describes it as Day 1.
- Figure 1: Was there a control to with just buffer to monitor the impact of injury (injury-induced mortality vs infection induced)? A more clearly defined injury control would help distinguish between immune-mediated and procedural mortality effects. Also, could variability in pathogen load confound the survival and immune response?
- Figures 3A and 4A: Do the points represent distinct replicates?
- Figures 5A and 5C present treatment comparisons, but it is not clearly stated in the figure captions what the controls are. Explicitly defining the control condition in each panel would reduce ambiguity. Also, the genes are listed only by LOC identifiers. Adding brief functional annotations would make the data more accessible and meaningful at first read.
- Figure 5F: What data is used for the PCA? Identified genes?
- Figure 5G: There appear to be multiple traces, but this isn't specified, and the colors are difficult to distinguish.
- Figure 5H: The legend describes this as Genome-wide distribution (line 386), but it seems to be mapping peaks onto a set of generalized features? This needs to be better explained.
- Figure 5L-M: The number of samples and biological replicates for ChIP-qPCR is not specified. Clarifying sample sizes is essential for interpreting reproducibility.
- Figure 6: The schematic includes information that wasn't presented elsewhere in the manuscript. For example, it labels the site of hemocytes as "ostia"

(Remarks on code availability)

N/A

Reviewer #4

(Remarks to the Author)

(Remarks on code availability)

Reviewer #5

(Remarks to the Author)

(Remarks on code availability)

Code for RNAseq experiments is working, but I cannot find code used for stats and plot of qPCR and metabolomics data.

Version 1:

Reviewer comments:

Reviewer #1

(Remarks to the Author)

The authors clarified comments and concerns I had with sample sizes and statistics. They also indicated they used one frame of brood sourced across one of several open-mated honey bee colonies. I see their data are also now available (or I just missed it last time, either way, thank you).

I have seen most of my concerns handled. But, knowing that they sourced the data from multiple colonies it would be worthwhile for them to either 1) include colony as a variable in their ANOVA and/or 2) demonstrate colony is not a significant contributor, generally.

(Remarks on code availability)

Reviewer #2

(Remarks to the Author)

The revised version of the manuscript entitled "Gut microbiota-derived butyrate primes systemic immunity in honey bees by mediating lipid metabolic reprogramming", by Liu et al., has been significantly improved by the authors, who have adequately addressed all the issues we raised on the original version.

(Remarks on code availability)

I was able to run the code, which is a usable resource for the community.

Reviewer #3

(Remarks to the Author)

We appreciate that the authors very thoroughly addressed all of our comments and concerns. The revised text clarifies the background and methods, and the additional analyses and experiments support the conclusions. Overall, the revised manuscript presents compelling and well supported findings.

(Remarks on code availability)

Reviewer #4

(Remarks to the Author)

(Remarks on code availability)

Reviewer #5

(Remarks to the Author)

(Remarks on code availability)

The code is working and provides useful resources for reproducing the results of the manuscript

Version 2:

Reviewer comments:

Reviewer #1

(Remarks to the Author)

The authors have handled all of my comments exceptionally and I see no further issues that would warrant another round of review.

(Remarks on code availability)

The code is available and interpretable

Point-to-point responses

Comments from Reviewer #1 (Remarks to the Author):

This study investigates how the honey bee gut microbiota, particularly the metabolite butyrate, primes systemic immune responses. Using germ-free bees and supplementation experiments, the authors show that butyrate triggers lipid metabolic reprogramming in the fat body. This leads to prostaglandin E2 (PGE2) synthesis and enhanced immune gene expression. The work reveals a novel microbiota–metabolite–immunity axis conserved across insects and mammals.

Impressive multi-omics integration (transcriptomics, metabolomics, CUT&Tag, qPCR); well-designed experimental contrasts (GF, HK, CV, CL); strong mechanistic narrative linking butyrate to PGE2 to immunity.

Response: We appreciate your clear and accurate summary of the key findings and novelty of our study. The manuscript has been revised accordingly to address the comments provided below.

What stats software was used and where are the codebases for statistical works?

Response: For analyses on RNA-seq and CUT&Tag, we used R. Relevant codes were uploaded to Github (https://github.com/Liujiaming2025/DataAnalysis_GFButyrate.git) and Figshare (<https://doi.org/10.6084/m9.figshare.30239623.v1>) for public access. All other data analyses were conducted using GraphPad Prism (v9.3.1). As Prism employs a graphical user interface, no script-based code was generated. Detailed statistical methods are described in the respective figure legends.

For comparisons between two independent groups, data were first assessed for normality and homogeneity of variances. If both assumptions were met, an unpaired Student's t-test was applied. If the data were normally distributed but variances were unequal, Welch's t-test was used. For non-normally distributed data, the Mann–Whitney test was employed.

For comparisons among multiple independent groups in the qPCR data, normality and variance equality were also evaluated. If normality and homogeneity of variances were satisfied, a one-way ANOVA followed by Tukey's post hoc test was performed. If data were normally distributed with unequal variances, Brown–Forsythe and Welch ANOVA tests were used. For non-normally distributed data, the Kruskal–Wallis test with Dunn's post hoc test was applied.

We have revised the statistical methods, providing more details in the Methods: “Statistical analyses were performed in GraphPad Prism (v9.3.1). Comparisons

between two groups used unpaired Student's *t*-tests when data met assumptions of normality and homogeneity of variance, Welch's *t*-tests when variances were unequal, and Mann–Whitney tests for non-normal distributions. For multiple-group comparisons, one-way ANOVA with Tukey's HSD post hoc test was applied when assumptions were satisfied. Brown–Forsythe and Welch ANOVA were used for normal data with unequal variances, and Kruskal–Wallis tests followed by Dunn's post hoc test were applied otherwise. Statistical significance was defined as $P < 0.05$." (Lines 1079-1088).

Line 78 and this section: other sections used dot plots and bar charts to show sample sizes (e.g. fig 2) this section did not and I found it tedious to find the sample sizes.

Response: We have revised all qPCR figures (Figs. 1c, 3d, 5a-b, and 5d), converting them to dot plots with accompanying bar charts to better illustrate sample sizes and data distribution.

Line 78 onwards: Any indication that these effects are consistent across multiple colonies/genotypes? Context on bee genetic background is missing. How many colonies? Were they MDIs or SDIs? How much colony-level variation explained your data, etc? I worry that with the smaller sample size (e.g. for gene expression) you're stats are not necessarily robust.

Response: We appreciate the reviewer's thoughtful comments and agree that these are critical considerations for interpreting our results. We have clarified the experimental design and addressed each concern in detail below.

1. Control for genetic background within experiments

To minimize confounding effects of genetic variation, all bees used within a given experiment, including both treatment and control groups, were collected from the same brood comb within a single hive. Workers emerging from the same comb are typically super-sisters, sharing the same maternal origin and, in most cases, the same paternal lineage, as they are usually laid by the queen within a short time frame. This sampling strategy ensures that compared groups have nearly identical genetic backgrounds, thereby effectively controlling for intra-colonial genetic variation, even though the MDI (multiple-drone insemination) or SDI (single-drone insemination) status of individual bees was not determined.

2. Colony consistency

Following editorial requirements (i.e., all experiments should be performed in at least two to three independent batches, or have at a minimum 5-10 independent biological replicates), we repeated the majority of our key experiments (including survival assays, hemocyte staining, butyrate and lipid droplet quantification, PGE₂ assays, HDAC activity measurements, and CUT&Tag-qPCR) in two additional independent

experimental batches using bees collected from different hives. This allowed us to examine colony consistency between experiments.

The difference in AMP expression between GF and CL bees was consistently significant between the two independent experiments (Figures 2c and 4g). Similarly, the expression of the ten genes involved in glycerolipid and arachidonic acid metabolism (GF vs. butyrate-treated) was also replicated in two trials (Figures 3c and 5a), where nine of the ten genes consistently show differential expressions.

3. Biological replicates and statistical design

Our experimental design incorporated biological replication at multiple levels to ensure statistical reliability. Twenty newly emerged bees under 24-hour age were randomly placed into each sterile plastic cup for bacterial inoculation or treatment. For each treatment, cup cages were established as biological replicates. Individual bees were randomly chosen from each cup cage for analysis.

Specific sample sizes for key assays were as follows:

qPCR (Figures 1c, 3c, 4g, 5a, 5c): $N = 6-8$ bees per group, randomly selected from three cup cages.

TAG measurement (Figure 3f): $N = 6$ biological replicates per group, each comprising fat body tissue pooled from 3 bees collected across three cup cages.

PGE2 measurement (Figure 4c, 4e): $N = 5-6$ biological replicates, each comprising tissue pooled from 5 bees randomly collected from three cup cages.

The reproducibility of results across independent experimental batches and biological replicates supports that the observed effects are not consequences of confounding factors related to colony effect or sample sizes.

We have added one sentence in the Methods section: “Within each experiment, bees for all treatment and control groups were derived from a single brood comb to control for genetic background.” (Lines 654-656)

All details regarding biological replicates and independent trials have been clarified in the relevant figure legends.

Lines 61-75: Consider making the last sentence more specific about the downstream applications or broader impact.

Response: We have modified the last sentence to: “These findings advance our understanding of the interplay between microbial metabolites and host immunity, and offer a foundation for designing targeted probiotic or metabolite-based strategies to improve pollinator health.” (Lines 89-92)

Line 132: Would be helpful to mention natural butyrate levels in bees for ecological relevance and to have a citation here.

Response: As suggested, we have added the natural butyrate levels in bees and the relevant literature in the revised version: “The microbial-derived SCFAs activate systemic immunity in mammals (Jordan & Clarke, 2024, doi: 10.1016/j.it.2023.12.002). In honey bees, acetate is most abundant in the hindgut of CV bees (>100 mM), while butyrate shows the highest level in hemolymph (22.8 mM) (Zheng et al., 2017, doi: 10.1073/pnas.1701819114).” (Lines 170-172).

Lines 189–225: It’s unclear if transcript-level differences translate into functional lipid mobilization; could add a metabolic flux analysis.

Response: We agree that linking gene expression changes to metabolic flux analysis would strengthen our findings in functional lipid metabolism. However, due to the significant technical challenges associated with performing stable isotope tracing and in vivo metabolic flux analysis in live honey bees, this approach is beyond the scope of the present study.

While we acknowledge this limitation, the observed gene expression changes are most likely indicative of functional lipid metabolism. The transcriptional changes are enriched in enzymes in lipid metabolism. Congruently, butyrate treatment led to a significant decrease in triacylglycerol level in the fat body (Figure 3e and 3f), physiological evidence indicating lipid mobilization.

Nevertheless, we have revised the text to tone down definitive causal claims, and acknowledged the lack of direct flux data as a limitation in the Discussion section. “..., our results suggested that honey bees utilize butyrate to upregulate TAG catabolic genes, promoting glycerate-to-DAG conversion...” (Lines 595-597)

We have also added “Future studies could directly track lipid metabolic fluxes in vivo, thus testing the model suggested by our transcriptomic data.” (Lines 598-600)

Lines 321–363: While I find you have a compelling dual mechanism (GPR41 and HDAC inhibition), isn't HDAC activity here only inferred from TSA and H3K27ac? Consider discussing limitations of specificity.

Response: Our data show that butyrate feeding upregulates both HDAC activity (Figure 5c) and H3K27ac levels (Figure 6a) in the fat body, leading us to infer a functional relationship between them. However, we acknowledge that direct genetic or pharmacological manipulations such as knocking down of the proposed causal genes can provide more conclusive evidence. To address this limitation, we have added the

following statement to the Discussion section: “To establish causality, a follow-up study could employ RNAi-mediated knockdown in combination with butyrate treatment, which would help determine the specific butyrate-regulated genes.” (Lines 616-618)

Reviewer #1 (Remarks on code availability):

The code and data sets are available but the code should be on something more permanent than github. I don't see the data yet, but they have provided accessions.

Response: Our code has been deposited on both GitHub (https://github.com/Liujiaming2025/DataAnalysis_GFButyrate.git) and is also available on Figshare, which partners with Nature Communications (<https://doi.org/10.6084/m9.figshare.30239623.v1>).

The code does also look to just be the alignment-based pieces. No statistical pieces have been included here.

Response: Statistical analyses were performed in GraphPad Prism (v9.3.1). As Prism utilizes a graphical user interface, no script-based code was generated for these analyses. Detailed statistical methods are described in the respective figure legends, and we have also updated the "Statistical analysis" section to provide additional details. (Lines 1079-1088)

Comments from Reviewer #2 (Remarks to the Author):

The manuscript titled “Gut microbiota-derived butyrate primes systemic immunity in honeybees by mediating lipid metabolic reprogramming” describes a regulatory mechanism unveiling a novel gut microbiota-immune-lipid metabolic axis for insects. The study is based on immune challenge under different microbiota backgrounds, followed by metabolomics and epigenomics analyses, offering an in-depth characterization of the molecular mechanisms underlying immune priming via gut in honeybees.

The study is accurate, the experimental methods sound and adequate. The results are well presented and provide interesting new information on insect immunity and, more in general on animal immunity. The data interpretation and discussion are adequate and require only some additional considerations as indicated in the comments below.

We have only a few suggestions that the authors may want to consider for revision.

Response: We thank the reviewer for the positive assessment of our work and the summary of its novelties. We have revised the paper according to your comments below.

-line 40: Please, check reference 15. This article shows that *Lactobacillus* strains trigger gut immune response, against gut opportunistic pathogens of honeybee. It seems apparently not related to systemic immunity. Instead, you may want to consider articles such as Kwong et al 2017, in which apidaecin expression in hemolymph is studied in response to gut microbiota (DOI: 10.1098/rsos.170003).

Response: We believe the confusion may stem from reference numbering. In the original manuscript, reference 15 corresponded to Kwong et al., 2017 (10.1098/rsos.170003). In the revised version, this reference has been retained, and a new citation (Horak et al., 2020, doi: 10.1098/rspb.2020.1184) has been added to address the role of symbionts in honey bee systemic immunity.

-line 83: “gut homogeny” should be “gut homogenate”?

Response: Changed.

-figure 1C: we suggest analyzing the data using 2-way anova + post-hoc test, using different letters to indicate differences.

Response: We have re-analyzed the data in Figure 1c and Figure 1f using two-way ANOVA followed by Tukey’s post-hoc test. The figure legend and the Results section

have been updated accordingly. Significant differences are now indicated by different letters.

The revised results are: “The temporal analysis of AMP expression revealed distinct patterns across groups and time points (Figure 1c, Supplementary Figure 1c). When comparing groups at each time point, significant upregulation relative to GF controls was observed at 3 hours post infection (hpi) for Apidaecin in the CV group, Defensin-1 in the HK group, and Hymenoptaecin in both the HK and CV groups. By 12 hpi, Apidaecin and Hymenoptaecin were elevated in the HK group, and by 24 hpi, Abaecin was specifically upregulated in the HK group.” (Lines 107-113).

Moreover, qPCR data should be plotted using GF at time 0 (eg. 3hours) as calibrator, to provide a dynamic picture of the time-course expression. In fact, in the present figure, which showed each time point separately, the AMPs expression curve of GF bees is unrealistically flat (ie. expression of AMPs can't be always 1).

Response: As suggested, we have re-calculated the relative expression values using GF at 3 h as the calibrator and plotted the temporal dynamics of AMP expression as line graphs in the new Supplementary Figure 1c. For the main figure, to maintain clarity in comparing the treatment groups (CV, HK, GF) at each individual time point, we have retained the normalization to the GF group within each corresponding time point. Supplementary Figure 1c illustrates time-course changes relative to GF at 3 h. The results show clear temporal dynamics.

We have added descriptions on temporal dynamics of AMP expression in the revised text: “Analysis of expression changes over time within each group (relative to 3 hpi levels) showed that Apidaecin in the HK group and Hymenoptaecin in the HK and CV groups were significantly upregulated as early as 12 hpi. By 24 hpi, Apidaecin, Defensin-1, and Hymenoptaecin were significantly upregulated in all three groups, and Abaecin was upregulated in the HK group.”(Lines 113-117)

line 129: “triggers systemic immunity priming” stands for “primes systemic immunity”?

Response: We have changed the phrasing to “primes systemic immunity” as suggested.

lines 130-131: “We applied 10 mM of acetate and butyrate to the bees to examine their role in increasing bee immune responses observed in this study”. This sentence sounds a bit "dropped" without any reference to previous studies, hypothesis or findings. Why acetate and butyrate? Did you identify those metabolites in bees with functional GM? Or this is just a hypothesis generated by findings in mammal models? A short introductory sentence addressing this issue is required.

Response: We have added introductory sentences at the beginning of the paragraph to explain the rationale. The decision to focus on SCFAs was based on their established role in activating immunity in mammals and insects. Among various SCFAs previously found in honey bees, acetate is the most abundant in the hindgut of CV bees, while butyrate shows the highest concentration in hemolymph. Therefore, we selected acetate and butyrate to investigate their effects on immunity.

The revised paragraph now begins with: “The microbial-derived SCFAs activate systemic immunity in mammals ²⁵. In honey bees, acetate is most abundant in the hindgut of CV bees (>100 mM), while butyrate shows the highest level in hemolymph (22.8 mM) ²⁶. These SCFAs have recognized immune roles: acetate activates gut immunity in *Drosophila* ⁹, and butyrate increases immune-related gene expression in honey bees ³⁵. ” (Lines 170-174).

-lines 497-499: The discussion section would benefit from few lines about the origin of butyrate. From figure 2b, it seems that all core species produce butyrate, although at different levels. Do they have the same metabolic capabilities to produce this metabolite? May butyrate be metabolized from dietary fibers, such as in other animals?

Response: We thank the reviewers for raising this point. After re-examining our statistical analysis, we found that the butyrate level in bees colonized with *Bombilactobacillus* (*Lactobacillus* Firm-4) was not significantly different from that in the GF group. We have corrected the figure and revised the corresponding text accordingly.

In honey bees, the primary substrates for butyrate production are likely complex polysaccharides, such as those contained in pollen cell walls, which could not be digested in host midgut, subsequently reaching the hindgut.

The microbial production of butyrate from dietary polysaccharides occurs via distinct pathways. Some bacteria directly break down polysaccharides into pyruvate and acetyl-CoA to generate butyrate. Alternatively, cross-feeding interactions involve initial fermentation of polysaccharides into intermediates (e.g., acetate, lactate) by some bacteria, which are then utilized by other specialist bacteria to produce butyrate (Vital et al., 2014, doi: 10.1128/mBio.00889-14). Four main pathways are known to contribute to butyrate biosynthesis: the acetyl-CoA, glutarate, 4-aminobutyrate, and lysine pathways (Vital et al., 2014, doi: 10.1128/mBio.00889-14).

Based on our measurement, all core species appear to be capable of producing butyrate except *Bombilactobacillus* (Figure 2b). But their role in butyrate production in the honey bee gut remains unclear. A detailed investigation into the genetic basis and pathway utilization of butyrate production in the bee gut microbiota represents an important direction for future research.

We have added one paragraph in the Discussion section to reflect these considerations: “In honey bees, butyrate is likely derived from the microbial fermentation of dietary polysaccharides from pollen cell walls that reach the hindgut. This process may occur via direct breakdown or cross-feeding among core species through known biochemical pathways⁵³, although the precise genetic determinants require further elucidation..” (Lines 560-564)

line 585: Indicate which model of microsyringe, which needle gauge and where bees were injected (e.g. between which abdominal segments? laterally, dorsally?).

Response: We have added a “Injection procedure” section in Methods: “All injection experiments were performed using 10 μ L microliter syringes (Shanghai Bolige Industry & Trade Co., Ltd.) fitted with needles of 0.26 mm in diameter. The injected volumes varied depending on the experiment and are specified in the respective Methods sections. In all cases, injections were performed manually through the central dorsal region of the thorax into the hemolymph of honey bees. The solution was delivered slowly over approximately 5 seconds to minimize tissue damage.” (Lines 710-717)

line 627: Which nanocarriers? Please, add this information

Response: We used the star polycation as the nanocarrier. We have added the information in the Methods. “The synthesized dsRNA was mixed with a star polycation nanocarrier in a 1:1 mass ratio to prevent degradation and improve delivery efficiency (Zhang et al., 2022, doi: 10.1021/acsami.2c08239).” (Lines 805-809)

Data availability: We appreciated the availability of RNAseq raw data and code, but we cannot find qPCR and metabolomic raw data + analysis/code.

Response: We have provided the raw qPCR as Supplementary Data. The metabolomic raw data have been uploaded to Metabolights, with the accession number MTBLS13098 (<https://www.ebi.ac.uk/metabolights/MTBLS13098>).

For the qPCR analysis, the $2^{-\Delta\Delta CT}$ method was used to normalize relative gene expression levels relative to the actin gene, as described in “Methods” of the original manuscript. The statistical analysis of qPCR data was performed using GraphPad Prism (v9.3.1). Since Prism is a graphical interface-driven software, it does not generate script-based code. We have updated the “Statistical Analysis” section (Lines 1079-1088) with further details. Visualization and plotting of the qPCR results were conducted in R using the ggplot2 package. The R code used for plotting has been included in GitHub (https://github.com/Liujiaming2025/DataAnalysis_GFButyrate.git) and is also

available on Figshare partnering with Nature Communications (<https://doi.org/10.6084/m9.figshare.30239623.v1>).

The metabolomics data were processed and analyzed on the Majorbio Cloud Platform (www.majorbio.com). Key analytical parameters have been included in the “Methods” section (Lines 1018-1031).

Code for RNAseq experiments is working, but I cannot find code used for stats and plot of qPCR and metabolomics data.

Response: Please see our response to the last comment.

Comments from Reviewer #3 (Remarks to the Author):

The manuscript "Gut microbiota-derived butyrate primes systemic immunity in honeybees by mediating lipid metabolic reprogramming" presents a promising study into how microbiota-derived metabolites, such as butyrate, modulate immune function in honeybees. The concept is timely and compelling, with potential relevance not only to bee studies but also to broader questions of host-microbe interactions and immune education. The authors tackle this through a multi-omics approach: microbiome manipulations using germ-free, heat-killed, conventionally colonized, coupled with targeted metabolite supplementation (e.g., butyrate and acetate) and metabolite quantification via LC-MS. They conducted pathogen challenge assays with the microbe *Hafnia* to assess immune competence and host survival. The study further integrated transcriptome sequencing and untargeted metabolomic analysis, qPCR for gene expression, assessment of lipid metabolism, and imaging of hemocyte aggregation amongst others, making the overall dataset rich and multidimensional. However, while the core idea is strong and the experimental breadth is commendable, several key issues need to be addressed to clarify data interpretation, strengthen the conclusions, and improve reproducibility. Below are comments detailing these concerns.

Response: We sincerely thank the reviewer for the comprehensive evaluation of our study and the detailed, constructive feedback. We have carefully revised the manuscript to address all comments accordingly.

Major Comments:

- Some of the data interpretation in the manuscript is confusing. A few examples of this include:
 - o Lines 91-94 imply that immune gene expression is significantly different in CV vs HK bees, while Figures 1C and S1B show that the relationship between these conditions are much more complex, and not significantly different across the board as implied.

Response: Reviewer #2 raised a similar question. We have re-analyzed the data in Figure 1c using two-way ANOVA followed by the Tukey's post-hoc test. The figure legend and results have been updated accordingly. The temporal analysis of AMP expression revealed distinct patterns across groups and time points (Figure 1c, Supplementary Figure 1c). For further details, please refer to our response to Reviewer #2. The corresponding result description has been updated in Lines 107-117.

The acetylsalicylic acid experiment is lacking the proper controls. In particular, there are no data shown for the effect of ASA treatment in a naïve bee. In particular, it needs to be demonstrated that the effect of ASA treatment is specifically decreasing survival post-infection because of the infection, rather than an indiscriminate impact on bee survival following ASA administration.

Response: Following this suggestion, we conducted an additional experiment to directly test the impact of ASA treatment on bee survival. Detailed methods are now included in the revised Methods: “To evaluate the potential toxicity of ASA, CL workers were fed sterile sucrose solution containing 10, 30, or 50 mM ASA (Cat#53526ES03, Yeasen, China) starting on day 3. Supplementation was maintained for seven days, and survival was recorded daily. Control bees received sucrose solution with an equivalent volume of PBS. Our results showed that 10 mM ASA supplement showed no impact on bee survival and this concentration was thus selected for subsequent experiments. CL bees (day 3) were fed 10 mM ASA for two days and sampled on day 5 for survival assays after *Hafnia* injection or RNA extraction and qPCR analysis.” (Lines 723-730)

The results showed that ASA alone is non-toxic to bees, but its presence abolishes the survival benefit conferred by microbial priming upon *Hafnia* challenge. This result demonstrates that the increased mortality stems from impaired PGE2-mediated immune protection, not from direct ASA toxicity.

The revised results now read: “CL honey bees were also treated with acetylsalicylic acid (ASA), a prostaglandin synthesis inhibitor. To examine whether ASA exerts toxic effects on the bees, bees were fed different concentrations of ASA without bacterial infection, whereas 10 mM ASA showed no significant effect on survival (Supplementary Figure 9c). However, following *Hafnia* infection, ASA-treated CL bees exhibited a pronounced reduction in survival, even lower than that of the GF group (Figure 4f).” (Lines 346-352)

The hemocyte area experiments are especially difficult to interpret:

The stained images are incredibly small and dark and don't appear to support the text descriptions. In particular, the DNA stain is very scattered, and doesn't seem to stain any of the proposed hemocytes.

Response: Hemocytes were stained with CM-DiI (red), GFP-*Hafnia* cells were visualized in green, and hemocyte nuclei were counterstained with Hoechst 33342 (blue). The staining experiments were independently repeated according to the previously described protocol (King & Hillyer, 2012, doi: 10.1371/journal.ppat.1003058).

The lack of clear visual distinction arises from the substantial size difference between hemocytes and their nuclei. In the main figure (Figure 1e; scale bar = 250 μ m), our goal was to illustrate the aggregation of hemocytes around the dorsal vessel. Because the nuclei are small and not easily resolved at this scale, the excitation intensity for Hoechst staining was inevitably reduced. However, higher-magnification images

(Supplementary Figure 1e; scale bar = 10 μm) show CM-DiI-labeled hemocytes actively phagocytosing GFP-*Hafnia*, with hemocyte nuclei clearly visible in blue. We have added a note in the revised Results section to guide readers to the corresponding appendix figure. “These hemocytes were engaged in phagocytosis of *Hafnia* (Supplementary Figure 1e).” (Lines 128-129)

"Hemocyte area" is a meaningless measure without more context (such as measures of cell density and cell size). Many insects have changes in cell number as well as the differentiation of distinct immune cells types following infection. Cell number and identity need to be determined directly.

Response: We agree that the term “hemocyte area” requires clearer definition. In the revised manuscript, we have changed it to “periostial hemocyte number” to more accurately reflect what the measurement represents.

This measurement is not intended as a proxy for total hemocyte counts for the insect, but rather specifically quantifies the recruitment and aggregation of hemocytes within the periostial regions of the heart, sites critical for pathogen clearance due to high hemolymph flow. The biological rationale for this focus has been elaborated in our response to the subsequent comment.

Methodologically, periostial hemocytes were labeled with CM-DiI, a dye validated for specific staining of phagocytic hemocytes in insects including honey bees (Hystad et al., 2017; doi: 10.1371/journal.pone.0184108; Yan & Hillyer, 2020; doi: 10.1126/sciadv.abb3164). We quantified the CM-DiI-positive area within a standardized field of view encompassing the periostial regions, which provides a reproducible and well-established proxy for hemocyte number at these immunologically active sites. This approach offers a biologically meaningful and functionally relevant measure of localized cellular immune activation. Further details regarding the cellular specificity of CM-DiI staining are provided in our response below.

We have revised the relevant sentence to: “At 3 hr post-infection, the periostial hemocyte number (quantified as the percentage of CM-DiI-positive fluorescent area relative to the total image area) in the CV group was approximately 2.6 times greater than that in the GF group and 2.4 times greater than that in the HK group (Figure 1f).” (Lines 129-132)

There is no context/description for why hemocyte aggregation would be a meaningful measure of immune competence. Is it a rough proxy for total immune cell numbers? Is this location somehow relevant to immune responses?

Response: We agree that the biological rationale for using hemocyte aggregation as a metric requires clarification. The insect dorsal vessel is anatomically divided into an

aorta in the thorax and a heart in the abdomen. The periostial regions surrounding the heart valves (ostia) are zones of intense hemolymph flow, making them critical sites for pathogen clearance (Hillyer, 2016, doi: 10.1016/j.dci.2015.12.006). Insect hemocytes exist in both sessile and circulating phases. Following infection, resident sessile hemocytes in the periostial regions become rapidly activated, while circulating hemocytes are recruited to these sites, resulting in localized aggregation and enhanced phagocytic activity (Hillyer & Pass, 2020, doi: 10.1146/annurev-ento-011019-025003). This periostial immune response represents a conserved and functionally significant defense mechanism across diverse insect taxa (Yan & Hillyer, 2020, doi: 10.1126/sciadv.abb3164).

The aggregation of hemocytes at periostial regions is not a proxy for total hemocyte numbers of the insect. Rather, it reflects a dynamic, infection-responsive process. The extent of periostial hemocyte aggregation has been quantitatively demonstrated to scale in a time- and dose-dependent manner with infection intensity (King & Hillyer, 2012, doi: 10.1371/journal.ppat.1003058). This established correlation validates the use of periostial hemocyte density as a specific and functionally meaningful indicator of cellular immune activation.

We have revised the manuscript to provide clearer descriptions of the biological rationale for using hemocyte aggregation: “In the insect circulatory system, the dorsal vessel comprises an aorta in the thorax and a heart in the abdomen. The periostial regions surrounding the heart valves (ostia) are zones of intense hemolymph flow, making them critical sites for pathogen clearance ³⁰. Insect hemocytes exist in both sessile and circulating states. Following infection, resident sessile hemocytes in periostial regions become activated while circulating hemocytes migrate to these regions, resulting in localized aggregation and enhanced phagocytosis ³¹. This periostial immune response represents a conserved and functionally significant defense mechanism across diverse insect taxa ³². The extent of periostial hemocyte aggregation has been quantitatively demonstrated to scale in a time- and dose-dependent manner with infection intensity ³³.”(see Lines 74–83).

Hemocytes are stained with CM-DiI. This has been shown to stain a class of hemocytes in mosquitoes, but there is no data or citation presented to validate that this stain is effective in bees. Particularly, it needs to be demonstrated that the CM-DiI signal is specific to hemocytes, and to determine which class(es) of hemocytes are stained by CM-DiI.

Response: CM-DiI is a lipophilic dye with high affinity to plasma membranes. It effectively labels insect hemocytes without staining fat body or other non-hemocyte tissues, and has been validated across a wide range of insect taxa, including honey bees (Yan & Hillyer, 2020, doi: 10.1126/sciadv.abb3164). In honey bees, CM-DiI has been reported to label plasmatocytes and granulocytes, both of which possess phagocytic

activity (Hystad et al., 2017, doi: 10.1371/journal.pone.0184108). In our study, we further confirmed the specificity of CM-DiI staining. At higher magnification (scale bar = 10 μ m), CM-DiI-labeled cells were observed actively phagocytosing GFP-tagged *Hafnia* (Supplementary Figure 1e), supporting that the stained cells are indeed functional hemocytes. Although this method cannot differentiate subtypes, this does not affect our conclusion as both types are involved in immunity responses (Hystad et al., 2017, doi: 10.1371/journal.pone.0184108).

We have added a few sentences to explain the CM-DiI staining in honey bee hemocytes. “Besides AMP expression, we quantified the conserved immune response of periostial hemocyte aggregation. This was done by measuring the periostial hemocyte number using CM-DiI, a well-established dye that specifically labels phagocytic hemocytes without staining non-hemocyte tissues, as validated in diverse insect species^{32,34}.” (Lines 122-125)

The experiments to test the role of butyrate are not definitive and do not support the conclusions drawn. The use of AR420626 and TSA do not explicitly test the effect that butyrate is having in bee immunity. They independently act in pathways previously linked to butyrate so are suggestive, but statements such as lines 334-335 "butyrate regulates lipid break-down in honeybees via both GPR41 activation and histone acetylation modulation" are not adequately supported by the data. More specific experiments are necessary to support this definitive conclusion.

At a minimum a more realistic and supportable statement of the findings is necessary. More direct experiments would considerably strengthen the conclusions of the manuscript – for instance, treating bees with butyrate + an appropriate GPCR receptor antagonist would allow for a stronger conclusion.

Response: To validate the involvement of GPR41 in mediating the effects of butyrate, we performed an additional experiment using β -hydroxybutyrate (SHB), which has been reported as an effective GPR41 antagonist in mammals (Kimura et al., 2011, doi: 10.1073/pnas.1016088108; Inoue et al., 2014, doi: 10.3389/fendo.2014.00081).

The details of the experimental design and methods are included in the revised Methods: “To investigate whether butyrate functions through GPCR signaling pathway, honey bees were treated with either a GPR41 antagonist, β -hydroxybutyric acid (SHB; Cat#300-85-6, MedChemExpress, USA), or a GPR41-specific agonist AR420626 (Cat#1798310-55-0, Cayman, USA).

For SHB treatment, SHB powder was dissolved in Ringer solution to prepare a 400 mM stock solution, which was then filtered through a 0.22 μ m membrane filter (Millipore Cat#SLGP033RB, Thermo Fisher Scientific, USA). To determine the appropriate supplementation concentration, SHB was added to 10 mM sodium butyrate-sucrose

solution to final concentrations of 20 mM, 50 mM, and 100 mM. GF and butyrate-supplemented bees received sucrose solution containing equivalent volumes of Ringer buffer. Newly emerged bees (< 24 h) were randomly assigned to groups and fed with the respective solutions for 7 days, during which survival was recorded daily. Based on the survival results, 20 mM and 50 mM SHB were selected for subsequent supplementation experiments.

For qPCR experiments involving SHB supplementation, GF and butyrate groups received sucrose solution supplemented with equivalent volumes of Ringer buffer. Newly emerged bees (<24 h) were randomly assigned to groups and fed with the respective solutions for 5 days. On day 5, abdominal carcasses were dissected for qPCR analysis.”(Line 914-932)

We found that the butyrate-induced upregulation of lipid metabolism-related genes was significantly suppressed when butyrate was co-administered with the GPR41 antagonist SHB (8 out of 9 genes). This result demonstrates that butyrate regulates gene expression primarily via the GPR41 pathway.

These new findings are updated in the new results: “To validate the regulatory role of butyrate through GPR41, GF honey bees were treated with butyrate alone, or in combination with β -hydroxybutyrate (SHB, a GPR41 antagonist) ^{45,46}. Potential toxicity of SHB was first assessed by feeding bees with varying concentrations of SHB. Results showed that neither 20 mM nor 50 mM SHB posed notable effects on bee survival (Supplementary Figure 12a). All ten lipid metabolism-related genes, but *Lpin*, were reproducibly up-regulated by butyrate (Figure 5a). This upregulation was significantly suppressed for eight genes when butyrate was co-administered with SHB (Figure 5a). Moreover, the inhibitory effect was more pronounced with 50 mM SHB than with 20 mM.” (Lines 412-421)

The methods are incompletely described, and so it would be difficult to replicate this work. The entire section needs to be improved, please see some specific examples below.

Response: We have thoroughly revised the Methods section according to your suggestions to ensure clarity and reproducibility.

The statistical analysis section is lacking in detail. Tests are listed, but in many instances, the test used in a specific experiment is not definitely stated, either in the methods section or the figure legends.

Response: Both Reviewer #1 and Reviewer #2 raised this question. Please refer to our response to the first comment from Reviewer #1. Detailed statistical methods are

described in the respective figure legends. We also update the statistical methods in detail in the “Statistical analysis” section of the Methods (Lines 1079-1088).

The authors should acknowledge the methodological and other limitations of the study for a more thorough discussion.

Response: As suggested, we have expanded our discussion to address limitations of the present study, including methodology, unresolved questions, and how these questions could be addressed in future studies. (Lines 547-552, 560-571, 598-600, 609-612, 616-618)

General/Minor Comments:

The introduction (e.g. lines 28-31) references immune priming in the context of establishing immune memory. The rest of the manuscript uses a distinct definition of "priming", which is closer to establishing immune competence. This aspect of the introduction should be rewritten to better reflect the findings of the manuscript.

Response: We agree that the term “immune priming” requires clarification to better reflect the context of our study.

The term “immune priming” in invertebrate immunology was first introduced by Little & Kraaijeveld (2004, doi: 10.1016/j.tree.2003.11.011). It describes a phenomenon where prior exposure to pathogens or microbial components enhances the immune response upon re-infection in invertebrate individuals or their offspring. While this effect is functionally analogous to immunological memory in vertebrates, it operates through distinct mechanisms (Sulek et al., 2021, doi: 10.1016/j.jip.2021.107656). The outcome of such priming is an elevated level of immune competence, enabling a stronger defense, which aligns with the operational definition used throughout our manuscript.

To ensure conceptual precision, we have modified the relevant sentences in the Introduction as follows: “Although insects lack acquired immunity, their innate immune system can develop enhanced defensive capacity through “immune priming”¹. This process enhances immune competence of the host after initial pathogen exposure, enabling more rapid and effective resistance and tolerance to subsequent infections¹⁻³. Besides pathogens and their microbe-associated molecular patterns, immune priming can also be induced by nonpathogenic bacteria or symbionts, stimulating the immune system to boost host defenses against subsequent pathogenic challenges^{3,4}.” (Lines 28-34)

We have also added in the Abstract: “The gut microbiota plays a crucial role in insect immune priming, inducing enhanced immune response that functionally resembles acquired immunity confined to vertebrates.” (Lines 11-13)

Lines 48-49 refer to bacteria species by outdated names (Firm-4 and Firm-5) which have been replaced by more descriptive names. Information is available in the Motta and Moran, 2023 review (citation 20).

Response: *Bombilactobacillus* and *Lactobacillus* have been used to replace Firm-4 and Firm-5 throughout the manuscript, as suggested.

Some of the writing is too specialized for publication in a general-interest journal. For example, line 63, refers to the pathogen *Hafnia* without giving a complete species name or any background information. Additionally, the section continues by discussing experimental treatments (lines 65-67) such as conventionally colonized, heat-killed, and germ-free, but without giving sufficient detail for someone outside the field to interpret the context of the experiments.

Response: We agree that the context and rationale for these experimental treatments were not sufficiently explained for a broader audience. For the CV, HK and GF groups, these setups were designed to disentangle the role of a live gut microbiota from other constitutional components of the bacteria. This allows us to specifically test the hypothesis that a live, metabolically active gut microbiota is required for enhanced survival upon infection. To address this, we have now added a detailed explanation in the Introduction section that clarifies the purpose of these distinct colonization methods. We have also added the species name and background of the pathogen *Hafnia alvei*.

The revised text now reads: “We challenged honey bees by injecting them with *Hafnia alvei*, a bacterium commonly detected in the hemolymph and guts of worker bees and known to cause septicemia and gut inflammation, thereby reducing overall survival^{10, 23, 28, 29}. Under *Hafnia* infection, we found that conventionally colonized bees (CV), which possess a complete and active gut microbiota, exhibited higher survival rates than both bees colonized with heat-killed microbiota (HK) and germ-free bees (GF) lacking gut bacteria. These results demonstrate that the metabolic activity of live gut microbes is essential for enhancing honey bee survival against pathogenic infection. The CV bees showed obvious upregulated AMP gene expression and dorsal vessel-associated hemocyte aggregation in the fat body, which represent humoral and cellular immune response respectively³⁰.” (Lines 63-73)

The study emphasizes butyrate as a central modulator of immune responses in honeybees. However, Line 68 raises the question of why butyrate was selected as the primary focus over other structurally similar metabolites. The authors briefly mention

acetate, but it would strengthen the rationale to elaborate on any prior evidence or biological plausibility supporting butyrate's unique systemic effects in bees.

Response: Reviewer #1 and Reviewer #2 also raised this point. Please refer to our response to Reviewer #1 and #2. We have added an introduction at the beginning of the paragraph to explain why we choose butyrate and acetate to test their immune roles (Lines 170-174)

Lines 98–100 (Figure 1): Is there a temporal resolution of immune activation? E.g. 24h GF compared to 3hr GF. 24h HK compared to 3h HK and is 12h CV higher than 3h CV? How does time post infection affect the hemocyte aggregation? How does 24h post infection have some level of aggregation (Figure 1F) yet the relative expression shown is very low.

Response: Reviewer #2 also raised the question about the temporal dynamics of AMP expression. As suggested, we have re-analyzed the data in Figure 1c using two-way ANOVA followed by the Tukey's post-hoc test. Both the figure legend and the Results section have been updated accordingly. For the AMP expression, we have updated the results in the revised version to include comparisons among groups at each time point and an analysis of expression changes over time within each group (relative to the levels at 3 hours post-infection) (Lines 113-117).

In accordance with the editor's suggestion, we conducted an additional independent experiment to analyze hemocyte aggregation (Figure 1f, Supplementary Figure 2-3). The data were analyzed using two-way ANOVA followed by Tukey's post-hoc test. The results indicated that within each group (relative to levels at 3 hpi), hemocyte aggregation was significantly downregulated in the CV group at 12 and 24 hpi, whereas no significant changes were observed in the GF and HK groups.

We have added one sentence to describe the temporal dynamics of hemocyte aggregation. "Hemocyte aggregation in the CV group was most prominent at 3-6 hpi and subsequently decreased, whereas no significant temporal changes occurred in the GF and HK groups (Figure 1f, Supplementary Fig. 2-3)." (Lines 132-135)

Line 161-163: There is a discrepancy between current findings and prior reports (e.g., Horak et al., 2020) on the immunogenicity of heat-killed bacteria which deserves discussion. This study states that heat-killed gut bacteria showed no impacts on hemocyte aggregation, with or without pathogen infection, whereas previous work has suggested HK *S. alvi* can trigger immune responses. The authors should discuss their ideas underlying this difference.

Response: We thank the reviewer for raising this point. We believe that the difference between our findings and those of Horak et al. (2020) reflects distinct aspects of the insect immune response rather than a discrepancy.

At the level of systemic immune protection, our results are consistent with those of Horak et al. As shown in Figure 1b, colonization with heat-killed (HK) mixed microbiota significantly improved honey bee survival upon subsequent challenge with the pathogen *Hafnia alvei*, aligning with the protective effect reported by Horak et al. for HK *S. alvi*. Although AMP expression was not measured following HK colonization alone, we assessed it after pathogen challenge. The observed upregulation of several AMP and immune-related genes (Figure 1c and Supplementary Figure 1c) further supports the protective role of HK microbiota in the immune response.

The difference noted by the reviewer may be explained by the following factors:

- (1) We focused on different aspects of the immune responses: We assessed cellular immunity (hemocyte aggregation), whereas Horak et al. focused on humoral immunity (AMP expression). The humoral and cellular immune responses may have been elicited by different molecular patterns or exhibit distinct activation thresholds.
- (2) Complexity of the microbiota: The immunostimulatory signals from the mixed HK microbiota (containing various Gram-positive and Gram-negative bacteria) used in our study may differ from those elicited by the single HK *S. alvi* strain used by Horak et al. A mixed microbial community could induce a more complex immunomodulatory environment, potentially leading to different response outcomes.

We have added a few sentences to discuss these differences in the Discussion part. “Interestingly, although bees colonized with HK bacteria did not exhibit any change in periostial hemocyte aggregation, their immune gene expression profiles were significantly altered upon pathogenic infection. This suggests that the activation signals or thresholds for cellular and humoral immunity in insects may be distinct. Although the temporal dynamics of these two immune responses were not entirely consistent, they act in concert in the systemic immune defense.” (Lines 547-552)

Line 173: Although the CL group shows high post-infection survival, the study does not directly test whether the effect is due to synergistic interactions among the five core microbes. The discussion would benefit from either a mechanistic hypothesis or additional data to explain this enhanced protection.

Response: We agree with the reviewer that our study does not distinguish between a synergistic effect and the action of a single species in the CL group. While previous studies have established that individual core members *Gilliamella* and *Snodgrassella*

can enhance host survival post-infection (Kwong et al., 2017, doi: 10.1098/rsos.170003; Horak et al., 2020, doi: 10.1098/rspb.2020.1184), the protective roles of other constituents in our mixture, including *Bifidobacterium*, *Lactobacillus*, and *Bombilactobacillus*, remain largely unexplored.

We have added a few sentences in the Discussion section. “Our findings demonstrate that a defined consortium of core gut bacteria can prime systemic immunity. While *Gilliamella* and *Snodgrassella* are known to enhance survival^{15,16}, the collective protection we observed may also originate from the contribution of other symbiotic species, and from potential synergies among them. Moreover, microbial metabolites other than short-chain fatty acids might also contribute to these immune-priming effects.” (Lines 566-571)

Line 330: The study suggests HDAC inhibition by butyrate and TSA but does not specify which HDAC enzymes/isoforms are affected. Given the functional divergence among HDAC classes, this detail could clarify the pathway specificity.

Response: We agree that identifying the specific classes or isoforms affected would help clarify the pathway specificity. In our study, total HDAC activity was measured using a commercial insect HDAC activity assay kit, which does not distinguish between specific HDAC isoforms.

In eukaryotes, HDACs are classified into four classes (Gregorette et al., 2004, doi: 10.1016/j.jmb.2004.02.006). Studies in mammals suggest that butyrate primarily inhibits Class I and Class IIa HDACs (Dokmanovic et al., 2007, doi: 10.1158/1541-7786.MCR-07-0324). Although homologous classes are also present in honey bees, isoform-specific antibodies or assays are not yet available to precisely determine which HDAC isoforms are targeted by butyrate in this model. We have now included a discussion of this point in the revised manuscript.

The added discussion: “Studies in mammals suggest that butyrate primarily inhibits Class I and Class IIa HDACs⁶¹. Although homologous classes are also present in honey bees, isoform-specific antibodies or assays are not yet available to precisely determine which HDAC isoforms are targeted by butyrate in honey bees.” (Lines 609-612)

Line 132 says that acetate and butyrate were chosen "in approximation of natural conditions" but there are no data presented or cited which show the presence or concentration of these metabolites in bee hemolymph.

Response: Both Reviewer #1 and Reviewer #2 raised the same question. Please refer to our response to Reviewer #1. As suggested, we have added the natural butyrate levels in bees and the relevant literature in the revised version. (Lines 170-174)

The context for the CL treatment group is not fully explained. It appears that in natural conditions, bees don't have equal amounts of the core microbiome species (e.g. Motta and Moran, citation 20). The results seem comparable to the presumably more natural CV treatment, but it was a bit confusing to follow without a full explanation.

Response: The gut microbiota of honey bees is dominated by 5-8 core bacterial lineages (Motta & Moran, 2024, doi: 10.1038/s41579-023-00990-3). In this study, we established a CL group inoculated with a consortium consisting of an equal amount of five bacterial strains representing the five core gut bacterial taxa, to avoid potential confounding factors present in the guts of nurse bees in natural settings, such as pathogens.

Natural bee guts exhibit substantial variation in bacterial composition (Ellegaard & Engel, 2019, doi: 10.1038/s41467-019-08303-0; Su et al., 2022, doi: 10.3389/fmicb.2022.934459; Luo et al., 2024, doi: 10.1016/j.celrep.2024.114408). Therefore, no single fixed bacterial mixture can fully represent the full spectrum of natural gut communities. In practice, a balanced mixture with equal amounts of the core bacteria species serves as a close approximation to natural conditions while minimizing potential pathogenic effects.

Our recent work (Han et al. 2024, doi: 10.1073/pnas.2405410121) constructed a same CL group using the same five strains and confirmed successful colonization of all five core bacteria in both CL and CV groups on day 6 via qPCR, using universal and genus-specific primer pairs. These results showed that the total absolute bacterial abundance and the abundances of the five core species in the CL group were comparable to those in the CV group.

In another study, colonization with a reconstituted community of 11 strains (from seven core species) in equal amounts resulted in slightly higher abundances for most species in CL bees compared to hive bees of the same age. Nevertheless, the relative community composition resembled that of nurse bee guts based on 16S rRNA profiling (Kešnerová et al., 2017, doi: 10.1371/journal.pbio.2003467). Both studies indicate that using core bacterial strains of honey bees can assemble a structured community resembling the native gut microbiota.

We have added one sentence in the revised version to give a better explanation of CL treatment. “Natural bee guts exhibit variation in bacterial composition^{36,37}. In practice, a balanced mixture with equal amounts of the core bacteria species serves as a close approximation to natural conditions while minimizing potential pathogenic effects^{27,38}. Specifically, the CL bees were inoculated with a consortium consisting of an equal amount of five bacterial strains representing the five core gut bacterial taxa (*Gilliamella*, *Snodgrassella*, *Bifidobacterium*, *Bombilactobacillus*, and *Lactobacillus*). The

abundances of each core bacterium in CL-treated bees were similar to those in the CV group²⁷.” (Lines 182-189)

Lines 263-264 make reference to the substrate for prostaglandin. Prostaglandin is a lipid compound rather than an enzyme, so it's unclear what "substrate" implies in this context.

Response: We have modified the sentences to be “Among fat body DEGs induced by butyrate supplementation, Pla2 is responsible for arachidonic acid production. This fatty acid is the precursor for prostaglandin biosynthesis⁴².” (Lines 333-335)

Line 264 is also confusing ... " RNAi of Pla2 reduced its expression" – what is "its" referring to. RNAi of Pla2 reduces Pla2 expression? Or one of the other molecules discussed in the preceding sentences?

Response: We have revised the sentence to: "RNAi-mediated knockdown of Pla2 reduced Pla2 expression." (Lines 335 in the revised version)

Confusing wording: "Cohesively" in line 330

Response: We have deleted the word “Cohesively”.

- Line 332 refers to expression effects on the "aforementioned genes", but the genes presented are not the same as the genes from the previous result (which I would assume is meant by aforementioned genes). The lists have considerable overlap, but are not identical as implied by the text.

Response: We have removed the word “aforementioned”

Methods comments:

Generally, clarify the number of biological replicates per group for each assay and trial. Indicate the number of independent experiments and the number of bees or cups used per group in each trial to ensure reproducibility and statistical robustness.

Response: As suggested, we have clarified the number of independent experiments and the number of bees or cups used per group in each trial in the figure legends.

Line 558 states that bees were fed ASA on day 3, but doesn't specify the length of exposure.

Response: The Methods section has been updated accordingly. Please refer to the response to the comment regarding the additional experiments on ASA toxicity evaluation (Lines 723-730)

Sequence accessions are not given for the genes of interest.

Response: We have added the corresponding NCBI sequence accession numbers for all genes of interest in the revised manuscript (see Supplemental information Table S2). These include *Actin* (NM_001185146.1), *Apidaecin* (NM_001011642.1), *Abaecin* (NM_001011617.1), *Defensin-1* (NM_001011616.2), *Hymenoptaecin* (NM_001011615.1), *Toll* (XM_026440067.1), *Cactus-1* (XM_006567107.2), *Cactus-2* (XM_394485.7), *Dorsal* (XM_006566997.3), *PGRP-LC* (XM_006565506.3), *Dredd* (XM_006570913.3), *Relish* (XM_026444179.1), *LOC408559* (XM_392104.6), *LOC551968* (XM_624350.5), *LOC409261* (XM_026445108.1), *mino* (XM_395192.7), *LOC724951* (XM_001120852.5), *LOC724995* (XM_001120897.5), *Lpin* (XM_393684.7), *LOC726880* (XM_026442631.1), *bbc* (XM_016916632.2), *Pla2* (NM_001011614.1).

Line 627 states that dsRNA was mixed with "nanocarriers" without naming/describing the nanocarrier used.

Response: We have revised the method to describe the nanocarrier. "The synthesized dsRNA was mixed with a star polycation nanocarriers in a 1:1 mass ratio to prevent degradation and improve delivery efficiency⁷⁷" (Lines 805-807)

Line 629: Does the dose of 10 µg refer to RNA alone? Or RNA + nanocarrier?

Response: The 10 µg dose refers to the amount of dsRNA only. The synthesized dsRNA was mixed with nanocarriers at a 1:1 mass ratio, and each bee received 10 µg of dsRNA in the injection. The mass of the nanocarrier is not included in this dose.

We have revised the manuscript to make this point explicit: "Each GF bee and butyrate-supplemented bee received an injection of 10 µg of dsRNA into the hemolymph on day 4. The dose refers to the dsRNA content only, excluding the nanocarrier mass." (Lines 807-809)

Line 650 should describe how the fluorescence images were analyzed. It fails to mention the measure used, how fluorescence intensity differences were normalized/controlled, etc.

Response: All fluorescence images were acquired using a Leica SP8 confocal microscope under identical optical settings. For each image, the gain was adjusted uniformly to avoid overexposure while minimizing background noise. We have added the details for image analysis in the revised version.

“For image analysis, red fluorescence channel images (CellTracker labeling at 552 nm) were imported into ImageJ v1.53a (NIH, Bethesda, USA), converted to 8-bit, and inverted. A threshold was applied based on the actual distribution of labeled hemocytes to select positive signals while excluding background fluorescence. The threshold was set according to consistent criteria across all samples to ensure reproducibility. The percentage of positive fluorescent area relative to the total image area was calculated as a measure of hemocyte density. Since this metric is based on relative fluorescent area rather than absolute intensity, the results are directly comparable across samples.”(Lines 830-838)

Include methods used to confirm sterility in GF bees. Was sterility verified via plating, PCR, or another microbial contamination check? Also, specify whether any antibiotics other than kanamycin were used for strain maintenance or selection for the other bacteria species.

Response: In our study, the sterility of GF bees was verified by qPCR absolute quantification as previously described (Kešnerová et al., 2017, doi: 10.1371/journal.pbio.2003467; Han et al. 2024, doi: 10.1073/pnas.2405410121). Briefly, a standard curve was constructed using a plasmid containing the target sequence amplified with universal bacterial 16S rRNA primers. Plasmid copy numbers were calculated from plasmid concentration (measured by Qubit) and fragment length, and ten-fold serial dilutions (10^8 - 10^4 copies) were used to generate the standard curve based on Ct values. DNA was extracted from bee hindguts using the CTAB method, and bacterial loads were quantified by qPCR with universal 16S rRNA primers. Bees with a total bacterial load of $<10^6$ cells were considered germ-free. Only GFP-expressing *Hafnia alvei* was maintained in LB medium with 100 $\mu\text{g}/\text{mL}$ kanamycin, and no other antibiotics were used for any bacterial strains.

We have added the details in the Methods section: “The sterility in GF bees was confirmed by absolute quantitative PCR with universal 16S rRNA primers as previously described^{27,38}.” (Lines 663-664)

Report the duration of bacterial culture (number of hours grown before use). "Overnight" is ambiguous.

Response: We have clarified the duration of bacterial culture in the Methods section. All bacterial strains were cultured for 16 h prior to use. The Methods section has been updated to replace the ambiguous term “overnight” with this precise growth duration.

Revised method:

“*Hafnia alvei* was cultured in Luria-Bertani (LB) medium at 35 °C for 16 hr. GFP-expressing *H. alvei* was grown in LB medium supplemented with 100 $\mu\text{g}/\text{mL}$

kanamycin at 35 °C for 16 hr. For core gut bacteria, *Gilliamella apicola* W8127 and *Snodgrassella alvi* H11 were cultured for 16 hr in Brain Heart Infusion (BHI) medium at 35 °C in a CO₂ incubator, while *Bifidobacterium asteroides* W8118 was cultured for 16 hr in TPY agar medium under the same conditions. *Bombilactobacillus* W8086 and *Lactobacillus* W8171 were cultured in MRS agar medium at 34 °C in a CO₂ incubator for 16 hr. Bacterial strains were revived by streaking onto agar medium. Single colonies were picked, enriched, and identified by 16S rRNA PCR amplification with universal primers, followed by Sanger sequencing and BLAST analysis against the NCBI database to confirm the bacterial species. After cultivation, all bacterial suspensions were washed and diluted with PBS to adjust the optical density at 600 nm (OD₆₀₀) to 1 prior to use.”(Lines 696-708)

Describe how OD₆₀₀ was measured (e.g., spectrophotometer model, and cuvette size).

Response: We have updated the Methods section to include the procedure for measuring bacterial cultures to an OD₆₀₀ of 1. “For adjusting bacterial cultures to OD₆₀₀ = 1, bacterial cultures were washed and diluted with PBS, and 200 μL of the suspension was transferred to a Corning Costar® 96-well cell culture plate. Optical density at 600 nm (OD₆₀₀) was measured using a Molecular Devices i3x multifunctional microplate reader. The PBS dilution was adjusted as needed to achieve an OD₆₀₀ of 1. OD₆₀₀ measurements were performed in triplicate to ensure accuracy.” (Lines 688-693)

Indicate how bacterial strain identity and purity were verified—such as by 16S rRNA sequencing, PCR with species-specific primers, or phenotypic methods?

Response: The bacterial strains preserved in the lab were first streaked onto culture medium. Single colonies were then picked and enriched, followed by PCR amplification of the 16S rRNA gene using universal primers. The amplified products were Sanger sequenced, and the resulting sequences were compared against the NCBI database using BLAST to confirm species identity. Purity was ensured by the observation of uniform colony morphology on agar plates and by the presence of a single band of the expected size in agarose gel electrophoresis, followed by a clear, unambiguous sequencing chromatogram.

We have added details in the Methods section, “Bacterial strains were revived by streaking onto agar medium. Single colonies were picked, enriched, and identified by 16S rRNA PCR amplification with universal primers, followed by Sanger sequencing and BLAST analysis against the NCBI database to confirm the bacterial species.”(Lines 703-708)

Specify (briefly) the role and concentration of any additives used (e.g., 2,6-diaminopimelic acid).

Response: We have updated the Methods section to specify the role of concentration of 2,6-diaminopimelic acid (DAP).

“*Escherichia coli* MFDpir (carrying a shuttle plasmid with KanR and GFP sequences) was mixed with the recipient *H. alvei* strain ZL01 at a 1:10 ratio. The mixture was incubated for 16 hr on Luria-Bertani (LB) medium supplemented with 2,6-diaminopimelic acid (DAP) at a final concentration of 0.3 mM, which is required for the growth of MFDpir.” (Lines 752-756)

Indicate when and for how long SCFAs were administered, and describe whether consumption was monitored (e.g., measuring syrup intake or visual confirmation).

Response: The details of SCFA feeding have been updated in the Methods section.

“To supplement SCFA, sodium acetate (Cat#A1070, Solarbio, China) or sodium butyrate (Cat#S9491, Solarbio, China) were first dissolved in sterile water to prepare stock solutions, which were then filtered through a 0.22 μm membrane filter (Millipore Cat# SLGP033RB, Thermo Fisher Scientific, USA). The stock solutions were added to the sucrose syrup to a final concentration of 10 mM when feeding the GF honey bees. Newly emerged bees were randomly assigned to groups within 24 hr of emergence, and feeding was lasted for 5 days. The control group received sucrose syrup containing an equal volume of PBS. Freshly prepared SCFA-containing sucrose solution was provided daily to maintain consistent intake, and consumption was monitored visually by observing the decrease in liquid level in each feeding tube.” (Lines 678-687)

Include the preparation steps for PGE2: initial concentration in ethanol, subsequent dilution series (e.g., 1:10 dil) and final working concentration so that the final ethanol content can be assessed

Response: We have updated the details of preparing steps of PGE2 as suggested.

“PGE2 (Cat#60810ES03, Yeasen, China) was first dissolved in absolute ethanol to prepare a stock solution at 6 $\mu\text{g}/\mu\text{L}$. The stock was then diluted with Ringer solution at a ratio of 1:11 (stock:buffer), yielding a final working concentration of 0.5 $\mu\text{g}/\mu\text{L}$. For control treatments, ethanol was diluted with Ringer solution at the same ratio to match the solvent content.” (Lines 732-736)

Clarify the injection site for PGE2 (e.g., thorax vs abdomen, specific tergite location).

Response: We have updated the Methods section to clarify the injection site: “They were then injected with 1 μL of the PGE2 solution (0.5 μg) into the hemolymph through the central dorsal region of the thorax using a microsyringe”. (Lines 738-740)

Describe the duration of anesthesia and how it was induced (e.g., ice exposure for 5 minutes).

Response: In our experiments, the bees were anesthetized by placing the cups containing the bees on ice for 10 minutes, until the bees were immobilized, prior to injection. We have updated the Methods section to include this detail.

Revised text in Methods:

“On day 5, GF bees were anesthetized by placing the cups containing the bees on ice for 10 minutes, until the bees were immobilized. They were then injected with ...”.
(Lines 736-738)

“Five days after inoculation, GF, HK, CV, CL, and SCFA-supplemented bees were briefly anesthetized on ice for 10 min and injected with” (Lines 760-762)

Provide injection parameters: volume of injected solution, needle gauge/type, and injection rate if known (e.g., $\mu\text{L}/\text{sec}$).

Response: Reviewer #2 also raised this question. Please refer to our response to Reviewer #2. We have added a description of the injection procedure in the Methods section (Lines 710-717).

Indicate whether any anticoagulants were used during hemolymph collection.

Response: No anticoagulants were used during hemolymph collection. To prevent oxidation and coagulation, hemolymph was rapidly collected using a 10 μL pipette and immediately transferred into pre-chilled tube on ice. After collection, the sample was immediately stored at $-80\text{ }^{\circ}\text{C}$. All procedures were performed under low-temperature conditions to preserve sample integrity.

Identify the type of homogenizer used and the specific lysis buffer (name and composition, if applicable)

Response: In our experiments, tissue homogenization was performed using a 1 mL glass homogenizer (Cat#YA0850, Solarbio, China), and this information has been added to the Methods section (“Tissue collection”) (Lines 778-779). For PGE2, TAG, and HDAC detection assays, PBS was used during homogenization, as described in the Methods section (Lines 877, 846 and 910). For protein extraction and nuclear extraction, the corresponding lysis buffers supplied with the commercial kits were used, and the catalog numbers of these kits are specified in the Methods section (e.g., Line 942, 1036).

General: The figure legends are ambiguous about the statistical tests, and give options for tests which may have been performed rather than directly stating which test led to the significance determinations shown in the figures.

Response: We have updated the statistical method for each figure in the figure legend.

General: Many of the figure legends don't describe the figure fully. For example, several of the hemocyte stain images have blue signal, but it is not specified in the legend as the red and green signals are.

Response: We have revised all relevant figure legends to include complete descriptions. We have updated the hemocyte stain figures. In the main figure (Figure 1e, scale bar = 250 μm), we aimed to illustrate the clustering of hemocytes around the dorsal vessel. Because the nuclei are small and difficult to visualize at this scale, the excitation intensity for Hoechst was reduced. Higher-magnification images in Supplementary Figure 1e (scale bar = 10 μm) show CM-DiI-labeled hemocytes phagocytosing GFP-*Hafnia*, with the hemocyte nuclei clearly visible in blue.

Figure 1A: The schematic is difficult to interpret. It implies that bees were sacrificed for gene expression following survival analysis, whereas the text implies these were separate experiments. The schematic also shows that inoculation happened at Day 0, but the legend describes it as Day 1.

Response: In our experiment, the survival assay, gene expression analysis, and confocal microscopy were conducted as independent experiments. We have revised the schematic diagram in Figure 1a to distinguish these experiments. For the gut bacterial inoculation experiment, the newly emerged bees were inoculated within one day after emergence. The schematic diagram has been updated and the figure legend has also been modified accordingly.

Figure 1: Was there a control to with just buffer to monitor the impact of injury (injury-induced mortality vs infection induced)? A more clearly defined injury control would help distinguish between immune-mediated and procedural mortality effects. Also, could variability in pathogen load confound the survival and immune response?

Response: To address the concern about whether an “injury + buffer” control was included, we have provided the corresponding control experiment in Supplementary Figure 1a. In this experiment, bees were injected with PBS solution (buffer control; with the same injection volume as the infection groups, 1 μL ; $n = 60$). During the 7-day observation period, only 2 bees died (2/60), resulting in a survival rate of 96.7%. These results indicate that the injection procedure itself caused negligible mortality and

therefore procedural injury can be excluded as a factor influencing the experimental results.

In all infection experiments, *Hafnia alvei* cultures were adjusted to an OD₆₀₀ of 1 using PBS, and 1 µL of the bacterial suspension was injected into each bee using a microliter syringe, ensuring precise and consistent initial pathogen inoculum across all individuals and replicates.

We acknowledge that variability in pathogen load could theoretically influence survival and immune responses. However, we did not monitor bacterial loads in individual bees during the course of infection. This decision was based on the primary scope of our study, which was designed to characterize the systemic immune response to a standardized microbial challenge, rather than to track inter-individual variation in pathogen dynamics.

Figures 3A and 4A: Do the points represent distinct replicates?

Response: Yes, all points shown in these PCA plots correspond to distinct biological replicates. We have revised the figure legend.

“Figure 3a: Principal Component Analysis of transcriptomes of fat body and hindgut of GF and butyrate-supplemented bees (5 days post-supplementation). Each point represents one biological replicate, with three replicates per group.”(Lines 291-294)

“Figure 4a: Principal Component Analysis of abdominal metabolomic profiles from GF and butyrate-supplemented bees at day 5 post-supplementation (n = 6 per group, each point represents one biological replicate). ”(Lines 375-377)

Figures 5A and 5C present treatment comparisons, but it is not clearly stated in the figure captions what the controls are. Explicitly defining the control condition in each panel would reduce ambiguity. Also, the genes are listed only by LOC identifiers. Adding brief functional annotations would make the data more accessible and meaningful at first read.

Response: We have written “GF bees were used as the control.” in the figure legend for Figures 5a and 5c.

To improve clarity, we have added brief functional annotations for all the genes at their first appearance in the manuscript, and also in figure legend of Figure 3c. These genes are now labeled with their standard gene symbols or LOC identifiers, with annotations provided in parentheses where available, including LOC408559 (retinal dehydrogenase 1), LOC551968 (aldose reductase), LOC409261 (glycerol kinase), mino (LOC411724, glycerol-3-phosphate acyltransferase mino), LOC724951 (1-acyl-sn-glycerol-3-

phosphate acyltransferase beta), LOC724995 (1-acyl-sn-glycerol-3-phosphate acyltransferase alpha), Lpin (LOC410201, phosphatidate phosphatase), LOC726880 (pancreatic lipase-related protein 2-like), bbc (LOC552795, choline/ethanolamine phosphotransferase), and Pla2 (LOC406141, phospholipase A2). For Figures 5a and 5c, the legend now references the annotations already provided in Figure 3c to avoid redundancy.

Figure 5F: What data is used for the PCA? Identified genes?

Response: The data used for principal component analysis (PCA) in Figure 5F (Figure 6b in the revised version) was derived from the peak counts of both the GF and butyrate groups in the CUT&Tag experiment. Specifically, we first use the R package DiffBind to count the reads of all samples and performed normalization, thereby obtaining normalized read counts for each sample at each peak. These normalized counts were then used as input for PCA analysis, which demonstrates the overall differences among samples from different treatment groups. In the subsequent downstream analysis, we used the ChIPseeker package to annotate the genes associated with these peaks. However, it should be noted that the PCA analysis itself was based on the CUT&Tag peak signal intensity, rather than on the gene expression levels.

We have revised the figure legend accordingly to make this point clearer.

“(B) Principal Component Analysis based on normalized CUT&Tag peak counts from the fat bodies of GF and butyrate-supplemented bees 5 days post-supplementation.”(Lines 514-515)

Figure 5G: There appear to be multiple traces, but this isn't specified, and the colors are difficult to distinguish.

Response: In Figure 5G (Figure 6c in the revised version), the blue traces represent H3K27ac peaks in the GF group, the red traces represent H3K27ac peaks in the butyrate-supplemented group, and the gray trace represents the IgG negative control. To improve clarity and visual distinction in the image, we uniformly increased the line thickness and adjusted the transparency for all groups. This adjustment makes the H3K27ac peaks more prominent while keeping the IgG trace visible. The legend has also been updated to clearly indicate the meaning of each trace:

“(C) Overlapping peaks of H3K27ac in GF (blue) and butyrate-supplemented (red) groups, with IgG (gray) as the negative control.”(Lines 516-517)

Figure 5H: The legend describes this as Genome-wide distribution (line 386), but it seems to be mapping peaks onto a set of generalized features? This needs to be better explained.

Response: The original figure legend described this analysis as a “genome-wide distribution”, which could be misleading. A more accurate description is “Distribution of H3K27ac peaks across genomic features”. As shown in Figure 5H (Figure 6d in the revised version), we quantified the distribution of peaks for each sample group across major genomic features, including promoter, exon, intron, intergenic region, 5’UTR and 3’UTR. This analysis was performed using ChIPseeker. We have updated the legend to better reflect the content of the figure.

The revised figure legend:

“(D) Distribution of H3K27ac peaks across genomic features (promoters, exons, introns, intergenic regions, 5’UTRs, and 3’UTRs) for each sample group.” (Lines 517-519)

Figure 5L-M: The number of samples and biological replicates for ChIP-qPCR is not specified. Clarifying sample sizes is essential for interpreting reproducibility.

Response: For the ChIP-qPCR experiments shown in Figure 5L-M (Figure 6h-i in the revised version), each biological replicate consisted of fat bodies pooled from 5 bees. Both the GF and butyrate-treated groups included 3 biological replicates, and the experiment was independently repeated twice (results from replicate in Supplementary Figure 15b). We have added this information to the figure legend to clarify sample sizes and replicates.

Revised figure legend:

“(H-I) ChIP-qPCR of H3K27ac at LOC724951 (H) and LOC724907 (I) promoters. Bars represent mean \pm s.e.m. $n = 3$ replicates for each group, with each replicate pooled from 5 individuals from 3 cup cages (two independent experiments, see Supplementary Figure 15b for replicate data). $P = 0.0044$ (H) and 0.0122 (I) from two-tailed Student’s t-test.” (Lines 522-526)

Figure 6: The schematic includes information that wasn't presented elsewhere in the manuscript. For example, it labels the site of hemocytes as "ostia"

Response: The circulation of hemolymph in insects is primarily driven by the dorsal vessel, which is structurally divided into an aorta in the thorax and a heart in the abdomen. Ostia are the valve-like openings in the insect heart with high hemolymph flow (Hillyer & Pass, 2020, doi: 10.1146/annurev-ento-011019-025003). They are critical immune sites where periostial hemocytes aggregate to efficiently phagocytose invading microbes. The ostia are annotated in Figure 1d, where this term is first introduced.

Reviewer #3 (Remarks on code availability):

N/A

Reviewer #4 (Remarks to the Author):

Reviewer #5 (Remarks to the Author):

Reviewer #5 (Remarks on code availability):

Code for RNAseq experiments is working, but I cannot find code used for stats and plot of qPCR and metabolomics data.

Response: Reviewer #2 raised the same question. Please refer to our response to Reviewer #2.

Point-to-point responses

Comments from Reviewer #1 (Remarks to the Author):

The authors clarified comments and concerns I had with sample sizes and statistics. They also indicated they used one frame of brood sourced across one of several open-mated honey bee colonies. I see their data are also now available (or I just missed it last time, either way, thank you).

I have seen most of my concerns handled. But, knowing that they sourced the data from multiple colonies it would be worthwhile for them to either 1) include colony as a variable in their ANOVA and/or 2) demonstrate colony is not a significant contributor, generally.

Response: We acknowledge the reviewer's concern about the possible colony effects associated with the use of multiple sample batches. These batches were intentionally incorporated to validate different mechanisms and to test the robustness and reproducibility of our results and conclusions.

Regarding the potential confounding effects of colony/honeybee genetic background on the studied factors, the field has increasingly recognized such influences in recent years. To address this, two established methodologies are commonly applied:

(1) Conducting independent experimental batches using bees from distinct colonies. Consistent results across batches indicate that the observed effects are not driven by specific genetic backgrounds (e.g., Motta et al. 2022, 10.7554/eLife.82595; Quinn et al. 2024, 10.1038/s41564-023-01572-y).

(2) Sampling from multiple colonies within a single experimental design and incorporating colony as a random effect in statistical models, allowing explicit evaluation and control of its potential impact (e.g., Liberti et al. 2024, 10.1128/mbio.01034-24; Powell et al., 2025, 10.1098/rsos.242151).

In our study, we initially used the first approach, which aligns with current standards in the field for controlling genetic background. We obtained consistent results from two independent experimental batches, each using bees from the same colony within a batch. The consistency across batches reinforced our conclusions. Here, in response to the reviewer's suggestion, we performed mixed-effects modeling, which confirmed that colony effects did not significantly influence the outcomes. Together, these analytical steps further strengthen the robustness and reliability of our conclusions.

Specifically, in our experiments, bees were sourced from an apiary containing over 50 colonies. Importantly, for each independent experimental batch, a single brood comb was randomly selected. All treatment and control groups within a given batch were derived from the same comb to control for genetic background. This design allows us to alleviate confounding factors introduced by colonies and attribute variations

observed in results to treatments. On the other hand, bees used in different experimental batches originated from different, randomly selected colonies (a summary of independent replicates is provided in Table R1). This design may introduce colony variation as a potential batch effect. In the meantime, experimental batches using randomly selected colonies also provides an opportunity to examine the robustness of the results. In the original manuscript, we presented and analyzed the data from each batch, while the results showed consistency between batches, confirming our core findings.

Additionally, following the reviewer's suggestion, we conducted new statistical analyses to assess the potential influence of colony-level (batch) variation. We applied Linear Mixed Models (LMMs) for continuous data (e.g., qPCR, enzyme activity) and Generalized Linear Mixed Models (GLMMs) for binomial survival data. This mixed-model framework is particularly well-suited to our experimental design, as it allows us to include colony (batch) as a random effect. This approach appropriately accounts for the non-independence of data points from the same colony and provides valid, generalizable inference for our fixed effect of interest (treatment), thereby directly addressing the suggestion to "include colony as a variable." This method is commonly used in similar honey bee studies to account for colony-level variation (e.g., Powell et al., 2025, 10.1098/rsos.242151).

Continuous variables were checked for residual normality and log₂-transformed when necessary. For post-hoc comparisons, estimated marginal means were calculated using the emmeans package in R, with *P* values adjusted by the Tukey method. Statistical significance was defined when *P* < 0.05.

Statistical analysis revealed significant differences in honey bee survival and hemocyte aggregation among the Conventional (CV), Heat-killed (HK), and Germ-free (GF) treatment groups (Tables R2-R3). Specifically, butyrate and prostaglandin E2 (PGE2) treatments significantly influenced survival and hemocyte aggregation (Tables R2-R3). The expression of immune-related genes was affected by gut microbiota, while butyrate treatment modulated the expression of lipid metabolism genes (Table R4). Further analysis indicated that the presence of butyrate significantly altered PGE2 abundance, histone deacetylase (HDAC) activity, and H3K27ac histone modification levels at target gene loci (Table R5). **Importantly, across all assays, the fixed effect of treatment remained statistically robust after accounting for batch as a random effect.** The direction and magnitude of these effects were fully consistent with the results from our separate analyses of each independent batch. Therefore, the results and conclusions of our work remain unchanged.

We have updated the Methods section to clarify the sampling procedure as follows:
"Within each experiment, bees for all treatment and control groups were derived from a single brood comb to control for genetic background. For each independent replicate

experiment, a new brood comb was randomly selected from the apiary containing more than 50 colonies.” (Lines 656–657)

Table R1. Experiments with independent batches

Experiment	Figures
Survival rate in GF, HK and CV groups	Fig. 1b, Fig. S1b
Survival rate in GF, CL, acetate-fed and butyrate-fed groups	Fig. 2a, Fig. S4b
Survival rate in GF, GF + PGE2 treated, CL, and CL+ASA treated groups	Fig. 4f, Fig. S9d
AMP gene expression in GF and CL groups	Fig. 2c, Fig. 4g
Expression of lipid metabolism related genes in GF and butyrate-fed groups	Fig. 3d, Fig. 5a
Hemocyte aggregation in GF, HK and CV groups	Fig. 1f, Fig. S3
Hemocyte aggregation in GF, HK, CL and butyrate-fed groups	Fig. S5-S6
Hemocyte aggregation in GF, GF + PGE2 treated, CL, and CL +ASA treated groups	Fig. S10-S11
PGE2 abundance in hemocyte and hindgut	Fig. 4c, Fig. S9b
HDAC activity in GF, butyrate-fed and TSA-treated groups	Fig. 5c, Fig. S12b
ChIP qPCR of H3K27ac at two genes	Fig. 6h-i, Fig. S15b

Table R2. Statistical analysis of survival rate with combined data from two independent batches

Experiment	Contrast	p.value
Survival rate in GF, HK and CV groups (Fig. 1b, Fig. S1b)	CV - GF	2.30E-14
	CV - HK	8.77E-08
	GF - HK	1.94E-06
Survival rate in GF, CL, acetate-fed and butyrate-fed groups (Fig. 2a, Fig. S4b)	Acetate - Butyrate	3.69E-14
	Acetate - CL	1.00E-15
	Acetate - GF	0.53
	Butyrate - CL	7.15E-03
	Butyrate - GF	3.70E-14
	CL - GF	1.00E-15
Survival rate in GF, GF+PGE2 treated, CL, and CL+ ASA treated groups (Fig. 4f, Fig. S9d)	GF - PGE2	7.09E-14
	CL - GF	4.75E-14
	ASA - CL	3.63E-14
	ASA - GF	2.52E-09
	ASA - PGE2	3.67E-14
	CL - PGE2	0.13

Table R3. Statistical analysis of hemocyte aggregation with combined data from two independent batches

Experiment	Contrast	Time post infection /Status	p.value	
Hemocyte aggregation in GF, HK and CV groups (Fig. 1f, Fig. S3)	CV - GF	3 h	4.40E-07	
	CV - HK		1.29E-07	
	GF - HK		0.96	
	Hemocyte aggregation in GF, HK and CV groups (Fig. 1f, Fig. S3)	CV - GF	6 h	1.64E-06
		CV - HK		2.50E-08
		GF - HK		0.59
		CV - GF	12 h	0.08
		CV - HK		0.06
		GF - HK		0.99
	Hemocyte aggregation in GF, HK and CV groups (Fig. 1f, Fig. S3)	CV - GF	24 h	0.11
		CV - HK		0.84
		GF - HK		0.31
Hemocyte aggregation in GF, HK, CL and butyrate-fed groups (Fig. S5-S6)		CL - GF	Pre-infection	3.35E-07
		GF - HK		0.70
		Butyrate - GF		1.63E-07
	Hemocyte aggregation in GF, HK, CL and butyrate-fed groups (Fig. S5-S6)	CL - GF	Post-infection	7.78E-09
		GF - HK		0.69
		Butyrate - GF		6.17E-08
Hemocyte aggregation in GF, GF+PGE2 treated, CL, and CL+ASA treated groups (Fig. S10-S11)	GF - GFPGE2	Pre-infection	4.83E-08	
	CL - GF		6.97E-07	
	CL - CLASA		1.65E-05	
	Hemocyte aggregation in GF, GF+PGE2 treated, CL, and CL+ASA treated groups (Fig. S10-S11)	GF - GFPGE2	Post-infection	4.29E-09
		CL - GF		1.89E-06
		CL - CLASA		7.85E-08

Table R4. Statistical analysis of gene expression with combined data from two independent batches

Experiment	Contrast	Status	Gene	p.value
AMP gene expression in GF and CL groups (Fig. 2C, Fig. 4g)	CL - GF	Pre-infection	Apidaecin	2.27E-07
			Abaecin	2.18E-03
			Defensin	0.73
			Hymenoptaecin	2.77E-04
		Post-infection	Apidaecin	3.25E-05
			Abaecin	2.95E-04
			Defensin	1.01E-03
			Hymenoptaecin	1.74E-04
			LOC408559	3.88E-05
			LOC551968	1.32E-08
Expression of lipid metabolism related genes in GF and butyrate-fed groups (Fig. 3d, Fig. 5a)	Butyrate - GF	-	LOC409261	1.59E-07
			mino	1.94E-06
			LOC724951	1.66E-05
			LOC724995	1.58E-08
			Lpin	1.84E-02
			LOC726880	2.11E-04
			bbc	4.67E-06
			Pla2	2.03E-05

Table R5. Statistical analysis of PGE2 abundance, HDAC activity and ChIP qPCR with combined data from two independent batches

Experiment	Contrast	Tissue/Gene	p.value
PGE2 abundance in hemocyte and hindgut (Fig. 4c, Fig. S9b)	Butyrate - GF		3.79E-05
	CL - GF	Hemolymph	1.11E-05
	CV - GF		4.89E-06
	Butyrate - GF		1.83E-05
	CL - GF	Hindgut	2.14E-05
	CV - GF		4.99E-05
HDAC activity in GF, butyrate-fed and TSA-treated groups (Fig. 5c, Fig. S12b)	Butyrate - GF		4.58E-08
	GF - TSA	-	3.84E-07
ChIP qPCR of H3K27ac at two genes (Fig. 6h-i, Fig. S15b)	Butyrate - GF	LOC724907	6.06E-04
		LOC724951	2.86E-04

Point-to-point responses

Reviewer #1 (Remarks to the Author):

The authors have handled all of my comments exceptionally and I see no further issues that would warrant another round of review.

Response: We thank the reviewer for the positive assessment of our work.

Reviewer #1 (Remarks on code availability):

The code is available and interpretable.

Response: Thank you for your positive note on code availability.